# Why and How LLMs Hallucinate: Connecting the Dots with Subsequence Associations

**Yiyou Sun[1], Yu Gai[1], Lijie Chen[1], Abhilasha Ravichander[2], Yejin Choi[3], Nouha Dziri,[4] Dawn Song[1]**

[1]University of California, Berkeley, [2]Max Planck Institute, [3]Stanford University, [4]AI2

## Abstract

Large language models (LLMs) frequently generate hallucinations—content that deviates from factual accuracy or provided context—posing challenges for diagnosis due to the complex interplay of underlying causes. This paper introduces a *subsequence association* framework to systematically trace and understand hallucinations. Our key insight is that hallucinations arise when dominant hallucinatory associations outweigh faithful ones. Through theoretical and empirical analyses, we demonstrate that decoder-only transformers effectively function as *subsequence embedding* models, with linear layers encoding input-output associations. We propose a tracing algorithm that identifies causal subsequences by analyzing hallucination probabilities across randomized input contexts. Experiments show our method outperforms standard attribution techniques in identifying hallucination causes and aligns with evidence from the model's training corpus. This work provides a unified perspective on hallucinations and a robust framework for their tracing and analysis. Our code is available at `https://github.com/sunyiyou/SAT.git`.

## 1 Introduction

Despite with LLM's widespread applications [64, 40, 26, 45, 76, 13], hallucination—a phenomenon where the model generates content that deviates from factual accuracy or intended meaning—remains a pervasive challenge. This issue not only undermines the reliability of LLMs but also limits their broader applicability. Unlike traditional software systems, where debugging mechanisms can trace errors to their source, hallucinations in LLMs lack analogous methods for causal analysis. When hallucinations occur, developers are left without effective tools to trace their origins, hindering their ability to understand and fix the problem.

Tracing the origins of hallucinations in language models is inherently challenging for several reasons. First, understanding their sources is difficult due to the multifaceted and entangled nature of the underlying causes. While previous research has provided valuable insights, it often focuses on isolated factors. These include, but are not limited to, overconfidence [66], decoding randomness [29], snowballing effects [69], long-tailed training samples [56], misleading alignment training [63], spurious correlations [32], exposure bias [7], the reversal curse [8], and context hijacking [23]. This complexity necessitates a unified framework that facilitates analysis while being both general and representative of the aforementioned causes.

Moreover, strategies for tracing hallucinations remain insufficiently explored. Traditional attribution methods, such as perturbation-based approaches [48, 52, 4], and gradient-based approaches [31, 57, 19], typically produce token-wise importance scores. However, these scores are limited in their utility, as they fail to establish causal relationships [2, 11, 53, 28, 9]. Attribution scores are also **confined to the immediate context**, while the same tokens, when presented in different contexts, may generate entirely different outputs. Consequently, the same hallucination phenomenon is often not reproducible, further complicating the task of identifying and addressing its sources.

These challenges raise a critical question: *What is the fundamental way to understand and trace hallucinations?* Addressing this requires an examination of the nature of decoder-only architectures, which are trained to map sequences of tokens of length $n$ from a vocabulary $\mathcal{V}$ to the next token.

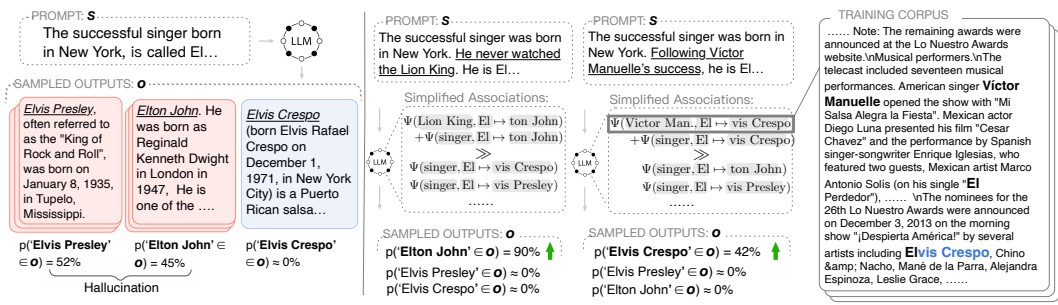

Figure 1: Hallucination examples and evidence of subsequence association in *Olmo-7B* [21]. (a) An example of a hallucination. The red blocks highlight hallucinated outputs with singers who were not born in New York. (b) Additional examples illustrating subsequence associations (using $\Psi$ as a measure). The probability of each singer's appearance changes with new sentences with special subsequence. These subsequence associations can be traced back to the training corpus, *Dolma* [54].

Given the astronomical number of possible token sequences, most of which appear only once in the training set [25, 54], it is infeasible for the model to verbatim learn all mappings. As a result, there are inevitably scenarios where the LLM heavily relies on a non-contiguous subset of tokens—referred to as a *subsequence*, which will lead to hallucinations when the subsequence represents an incomplete or incorrect context. This claim aligns with prior observations [20, 67].

For example, as shown in Figure 1(a), the LLM prioritizes the subsequence containing "successful singer" and "El" while ignoring "born in New York," leading to an input-conflicting hallucination [72]. Our key insight is that these subsequences act as "triggers" in token prediction, with their associations to target tokens embedded in the model. Further evidence, shown in Figure 1(b), demonstrates that adding a sentence like "he never watched the Lion King" increases the likelihood of generating "Elton John," who composed songs for the movie.

Within the framework of subsequence association, we introduce algorithms in Section 3 to trace causal subsequences that lead to hallucinations. Rather than relying on token-wise importance scores confined to a single input context, our method generates a diverse corpus of sentences that randomly incorporate subsequences from the original prompt. By analyzing these input–output pairs, we measure the conditional probability of hallucination tokens appearing given each subsequence, thereby identifying triggers that reproducibly induce the same hallucination across varied contexts. This focus on **cross-context reproducibility**—rather than minimizing a prediction gap within one context—enables robust causal tracing of hallucination sources.

To highlight the significance and practicality of studying hallucinations through subsequence associations, this paper makes the following contributions:

- **Theoretical Foundation**: We provide a theoretical foundation in Section 2.3 showing that feature blocks in popular decoder-only transformer architectures can fundamentally function as subsequence embedding models, along with linear layers encoding input-output associations. Additionally, we demonstrate in Section 2.4 that a low error rate in subsequence association probabilities between the training data and the model's generation—particularly for high-frequency subsequences—is a sufficient condition for convergence.

- **Unifying Hallucination Causes**: We intuitively show in Section 2.2 how common hallucination sources identified in prior work can be unified under the subsequence association framework, offering a cohesive perspective for understanding and addressing hallucinations.

- **Reproducibility-Focused Attribution**: We introduce in Section 3 a reproducibility-centric metric that identifies subsequences consistently triggering hallucinations across diverse input contexts, thus overcoming the context-bound limitations of traditional attribution methods.

- **Empirical Insights**: We empirically validate in Section 4 our proposed subsequence association algorithm, demonstrating that it identifies more causal subsequences than traditional attribution methods. Our results also reveal that both small and large LLMs produce hallucinations that are largely influenced by specific subsequences. Additionally, the identified subsequences are supported by their association probabilities in the LLM's training corpus.

## 2 Foundation of Subsequence Association

In this section, we address three key questions: a) (Section 2.2) How does subsequence association contribute to understanding and tracing hallucinations? b) (Section 2.3) How does a decoder-only transformer encode subsequences? c) (Section 2.4) How is subsequence association learned during pre-training? We begin by introducing the basic notations and foundational setup.

**Setting.** A decoder-only large language model (LLM), denoted as $F(\cdot)$, maps an input sequence $\mathbf{s} = [x_1, x_2, \ldots, x_n] \in \mathcal{V}^n$, where $x_i$ represents individual tokens and $\mathcal{V}$ is the token vocabulary, to an output sequence $\mathbf{o} = [y_1, y_2, \ldots, y_m] \in \mathcal{V}^m$, where $y_i$ denotes generated tokens. Here, $n$ and $m$ represent the maximum allowable lengths for input and output sequences, respectively. For sequences shorter than these maximum lengths, padding tokens are added by default.

To simplify the sequence generation model $F(\cdot)$, we reduce it to a next-token prediction model, $F_{\text{next}}(\cdot)$, which is trained on a dataset $\mathcal{D} = \{(\mathbf{s}^{(i)}, y^{(i)})\}_{i=1}^N$. Each data point in $\mathcal{D}$ corresponds to a next-token prediction task. For instance, a document with $t$ tokens is transformed into $t - 1$ training data points, as each token (except the last) serves as input to predict the subsequent token.

The next-token predictor $F_{\text{next}}(\mathbf{s})$ outputs a token sampled from the categorical distribution $\text{Cat}(\hat{p}(y|\mathbf{s}))$, where

$$\hat{p}(y|\mathbf{s}) = \exp(f(\mathbf{s}, y)) / \sum_{y' \in \mathcal{V}} \exp(f(\mathbf{s}, y')).$$

Here, $f(\mathbf{s}, y)$ represents the logit score assigned to token $y$ given the input sequence $\mathbf{s}$. The model $F(\mathbf{s})$ is optimized by minimizing the negative log-likelihood loss:

$$\mathcal{L}(\boldsymbol{\theta}) = \mathbb{E}_{(\mathbf{s}, y) \in \mathcal{D}}[-\log \hat{p}(y \mid \mathbf{s})].$$

Achieving near-zero loss for a de-duplicated training dataset requires that the parameter size scales as $O(N)$, which is often unattainable in practice[1]. It highlights the necessity of studying subsequence associations as a means to better understand and mitigate hallucination phenomena.

### 2.1 Basics of Subsequence Associations

To facilitate analysis of subsequences, we first establish formal definitions to describe subset relationships in sequences.

**Definition 2.1 (Subsequence Relation ($\sqsubseteq$)).** Given two sequences $\tilde{\mathbf{s}}$ and $\mathbf{s}$, we say that $\tilde{\mathbf{s}} \sqsubseteq \mathbf{s}$ if and only if $\tilde{\mathbf{s}}$ can be derived by deleting zero or more elements from $\mathbf{s}$ while preserving the order of the remaining elements. The subsequence *does not need to appear contiguously*[2] in the original sequence.

Similarly, we use $\tilde{\mathbf{o}} \sqsubseteq \mathbf{o}$ to denote the subsequence in the output. To isolate and analyze hallucinations, we denote $\tilde{\mathbf{o}}^h \sqsubseteq \mathbf{o}$ as the **hallucinated subsequence**. For example, consider the response illustrated in Figure 1:

> *"The successful singer was born in New York. He never watched the Lion King. He is El​ton John."*

Here, $\tilde{\mathbf{o}}^h$ corresponds to the hallucinated token subsequence (ton, John). In this scenario, the subsequence (Lion, King, El) from the input may act as a causal trigger for generating the output $\tilde{\mathbf{o}}^h$. This connection arises because Elton John composed music for *The Lion King* movie. In this paper, $\tilde{\mathbf{s}} \sqsubseteq \mathbf{s}$ is often used to denote the subsequence of the input sequence that triggers the hallucinated subsequence $\tilde{\mathbf{o}}^h \sqsubseteq \mathbf{o}$. This triggering relationship is formally defined as follows:

**Definition 2.2 (Subsequence Association).** For input $\mathbf{s}$ sampled in the real-world input distribution $\mathcal{P}_{\mathbf{s}}$, the association between a subsequence $\tilde{\mathbf{o}}$ in the output $\mathbf{o} \sim F(\mathbf{s})$ and a potential triggering subsequence $\tilde{\mathbf{s}}$ in the input $\mathbf{s}$ is defined as:

$$\Psi(\tilde{\mathbf{o}}, \tilde{\mathbf{s}}) = \log \frac{\mathbb{P}_{\mathbf{s} \sim \mathcal{P}_{\mathbf{s}}, \mathbf{o} \sim F(\mathbf{s})} \left(\tilde{\mathbf{o}} \sqsubseteq \mathbf{o} \mid \tilde{\mathbf{s}} \sqsubseteq \mathbf{s}\right)}{\mathbb{P}_{\mathbf{s} \sim \mathcal{P}_{\mathbf{s}}, \mathbf{o} \sim F(\mathbf{s})} \left(\tilde{\mathbf{o}} \sqsubseteq \mathbf{o}\right)}.$$

---

[1] For instance, during the pretraining phase of Olmo-7B [21], the Dolmo dataset [54] used contains approximately 3 trillion tokens—far exceeding the model's parameter size.

[2] These subsequences are non-contiguous because LLMs often generate lengthy responses where hallucinated content is interspersed with normal content.

This measure extends the concept of *pointwise mutual information (PMI)*, originally introduced to quantify the association between individual tokens or words [14], to subsequences. While the exact computation of $\Psi(\tilde{\mathbf{o}}, \tilde{\mathbf{s}})$ is intractable due to the impossibility of enumerating all possible input sequences, the definition serves as a general notion in analysis. In practice, an approximate form will be employed empirically as detailed in Section 3.

## 2.2 Unified Understanding for Hallucination with Subsequence Associations Analysis

In this section, we explore how subsequence associations can be utilized to understand and analyze the hallucination phenomenon in a unified way. The goal in this section is to **provide an intuitive perspective** towards hallucination instead of a rigorous proof. To facilitate this analysis, we build upon the definition introduced earlier and present the following proposition (proof in Appendix B.4):

**Proposition 2.3.** *(Simplified form) Under the independence assumption for subsequences appearing in $\mathbf{s}'$, we have:*

$$\log \mathbb{P}(\tilde{\mathbf{o}} \sqsubseteq \mathbf{o}|\mathbf{s} = \mathbf{s}') = \sum_{\tilde{\mathbf{s}} \sqsubseteq \mathbf{s}'} \Psi(\tilde{\mathbf{o}}, \tilde{\mathbf{s}}) + \log \mathbb{P}(\tilde{\mathbf{o}} \sqsubseteq \mathbf{o}),$$

where we abbreviate the subscription $\mathbf{s} \sim \mathcal{P}_\mathbf{s}, \mathbf{o} \sim F(\mathbf{s})$ for simplicity. This proposition indicates that the probability of a hallucination subsequence $\tilde{\mathbf{o}}^h$ appearing in the output of $\mathbf{s}'$ is influenced by the cumulative effect of all subsequence associations. In practice, due to the language diversity, most subsequences have negligible associations with hallucinations, allowing us to narrow the analysis to dominant subsequences, as discussed next.

To formalize, we consider a faithful version $\tilde{\mathbf{o}}^g$ of the hallucinated output $\tilde{\mathbf{o}}^h$. For instance, in Figure 1, $\tilde{\mathbf{o}}^h$ is "ton John," while $\tilde{\mathbf{o}}^g$ is the desired "vis Crespo." These outputs are mu-

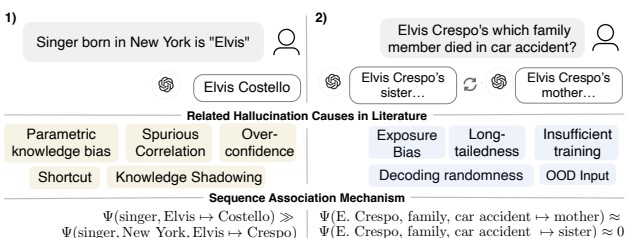

Figure 2: Examples of two hallucination cases of *GPT-4o* with simplified input/output provided here. Full screenshots are included in Appendix D. The right side of the examples displays the results of two generations. The faithful answers to the examples are *Elvis Crespo* (left) and his niece (right) respectively.

tually exclusive[3], as they typically correspond to predictions at the same token position. This leads to the central questions: a) *Why do the hallucinated subsequences $\tilde{\mathbf{o}}^h$ appear?* and b) *Why do the faithful ones $\tilde{\mathbf{o}}^g$ fail to appear?*

Since $\tilde{\mathbf{o}}^h \sqsubseteq \mathbf{o}$ and $\tilde{\mathbf{o}}^g \sqsubseteq \mathbf{o}$ are mutually exclusive events, hallucination occurs when:

$$\mathbb{P}(\tilde{\mathbf{o}}^h \sqsubseteq \mathbf{o}|\mathbf{s} = \mathbf{s}') > \mathbb{P}(\tilde{\mathbf{o}}^g \sqsubseteq \mathbf{o}|\mathbf{s} = \mathbf{s}').$$

By applying Proposition 2.3 and assuming, for simplicity, that the two events have approximately equal marginal probabilities $\mathbb{P}(\tilde{\mathbf{o}}^h \sqsubseteq \mathbf{o}) \approx \mathbb{P}(\tilde{\mathbf{o}}^g \sqsubseteq \mathbf{o})$, this inequality reduces to analyzing the relative strengths of subsequence associations: $\sum_{\tilde{\mathbf{s}} \sqsubseteq \mathbf{s}} \Psi(\tilde{\mathbf{o}}^h, \tilde{\mathbf{s}}) > \sum_{\tilde{\mathbf{s}} \sqsubseteq \mathbf{s}} \Psi(\tilde{\mathbf{o}}^g, \tilde{\mathbf{s}})$. In practice, hallucination often arises when a single trigger subsequence $\tilde{\mathbf{s}}$ dominates the associations. In this scenario, it allows us to simplify the analysis further:

$$\Psi(\tilde{\mathbf{o}}^h, \tilde{\mathbf{s}}^h) > \Psi(\tilde{\mathbf{o}}^g, \tilde{\mathbf{s}}^g),$$

where $\tilde{\mathbf{s}}^h$ and $\tilde{\mathbf{s}}^g$ are the subsequences with the strongest associations to $\tilde{\mathbf{o}}^h$ and $\tilde{\mathbf{o}}^g$, respectively. Even when hallucinations occur, the magnitudes of the associations $\Psi(\tilde{\mathbf{o}}^h, \tilde{\mathbf{s}}^h)$ and $\Psi(\tilde{\mathbf{o}}^g, \tilde{\mathbf{s}}^g)$ provide insights into the underlying causes. We present two typical cases below:

- $\Psi(\tilde{\mathbf{o}}^h, \tilde{\mathbf{s}}^h) \gg \Psi(\tilde{\mathbf{o}}^g, \tilde{\mathbf{s}}^g)$: In this scenario, the model produces hallucinations results with **over-confidence** [66]. It occurs when the language model exhibits a strong bias toward an incorrect or a limited subset of trigger subsequences in the input. This bias underlies phenomena such as **parametric knowledge bias** [46], **spurious correlations** [32], **shortcuts** [67], and **overshadowing** [73]. For instance, in the Figure 2 (left), *GPT-4o* relies on a shorter subsequence that is more prevalent in the training set, leading to hallucination.

---

[3]It is important to note that this analysis considers a single hallucination pathway for simplicity. In reality, multiple $\tilde{\mathbf{o}}^g$ ($\{\tilde{\mathbf{o}}_1^g, \tilde{\mathbf{o}}_2^g,...\}$) may compete with $\tilde{\mathbf{o}}^h$ simultaneously.

- $\Psi(\tilde{\mathbf{o}}^h, \tilde{\mathbf{s}}^h) \approx \Psi(\tilde{\mathbf{o}}^g, \tilde{\mathbf{s}}^g)$: Here, hallucinations arise due to inherent **decoding randomness** [29]. A typical sub-case is when the association magnitudes are close to zero, it indicates that **no** subsequence is strongly associated with both the hallucinated or faithful answer. This situation typically occurs when the input is **out-of-distribution** [36] or inconsistent with the training distribution (**exposure bias**) [7]. Other contributing factors include entities in the **long-tailed** [56] region, the model being **architecturally limited** [6], or **insufficient training** [72] with many factoids appearing only once [25]. For example, in the right example of Figure 2, *GPT-4o* encounters a question about a very rare fact and struggles to generate the correct answer.

Beyond these two primary cases, other causes of hallucination merit discussion: a) The **reversal curse** [8] occurs because the learned sequence associations in language models are unidirectional. Consequently, querying knowledge with reversed associations leads to suboptimal performance; b) The **snowballing effect** [69] or **context hijacking** [23] happens when the model is provided with a false premise or receives repetitive incorrect answers, strengthening the subsequence association to the hallucinated outcomes. For instance, in Figure 1, the mention of "Lion King" strengthens the association with "Elton John."; c) **Outdated or false knowledge** in the training dataset [18, 42, 56] lead to hallucinations by establishing incorrect subsequence associations.

> **Takeaway**
>
> Cases discussed in this section demonstrate how subsequence association analysis provides a unified framework for understanding common causes of hallucinations in language models.

## 2.3 Subsequence Associations Encoded in the Transformer Block

Building on the high-level intuition of how subsequence associations influence the behavior of hallucination, we now delve into the mechanisms by which decoder-only transformers encode these associations in their parameters. This discussion consists of two main parts: a) *Subsequence embedding in transformer feature blocks.* We demonstrate in Theorem 2.4 that transformer feature blocks can near-orthogonally embed each unique subsequence. b) *Subsequence association in the final feature block.* We illustrate how the final feature block encodes subsequence associations by mapping subsequence embeddings to output logits.

**Subsequence embedded in the transformer block.** The central takeaway from Theorem 2.4 is that the transformer feature block at the $t$-th position has the capacity to generate a feature that acts as a linear combination of embeddings of subsequences ending in the token $x_t$. This is illustrated in Figure 3, where a representative subsequence ending in the token El is shown. Due to the model's capacity limits, the number of subsequences a transformer can encode is constrained. Theorem B.3 in the Appendix provides a formal statement; a simplified version is presented here.

**Theorem 2.4** (Subsequence Embedding with Transformer). *(informal) For an input sequence* $\mathbf{s} = [x_1, x_2, ..., x_n] \in \mathcal{V}^n$ *and a subsequence set* $\mathcal{S}$ *with subsequence's length smaller than* $2^l$, *the feature in the $l$-th layer of transformer at $t$-th position can serve as the linear combination of the embeddings of the subsequences with the*

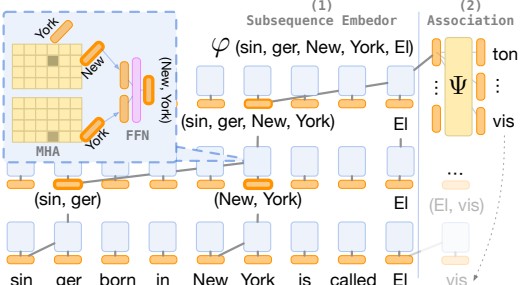

Figure 3: Illustration of subsequence embedding procedure via transformer blocks and the encoding of associations within matrix parameters. Orange boxes represent embeddings for individual tokens and subsequences. Blue boxes denote transformer blocks, each comprising Multi-Head Attention (MHA) and Feed-Forward Networks (FFN). The weight matrix is highlighted in yellow boxes. Grey lines depict the flow of token information used to form the subsequence embedding for (sin, ger, New, York, El).

*same ending token $x_t$ ($t$-token in $\mathbf{s}$). Specifically, the feature in the $l$-th layer is written as:* $\Phi^{(l)}(\mathbf{s}) = [\phi_1^{(l)}(\mathbf{s}), \phi_2^{(l)}(\mathbf{s}), ..., \phi_n^{(l)}(\mathbf{s})]^\top$, *where*

$$\phi_t^{(l)}(\mathbf{s}) = \sum_{\tilde{\mathbf{s}} \in \mathcal{S} \cap Sub(\mathbf{s},t)} \mu_{\tilde{\mathbf{s}}}^l \cdot \varphi(\tilde{\mathbf{s}}),$$

with coefficient $\mu_{\tilde{s}}^l > 0$ and $Sub(\mathbf{s}, t) = \{\tilde{\mathbf{s}} \mid \tilde{\mathbf{s}} \sqsubseteq \mathbf{s}_{[:t]}, \tilde{\mathbf{s}}_{[-1]} = x_t\}$ is a collection of the subsequences of the first-$t$-token sequence $\mathbf{s}_{[:t]}$ with the same ending token $x_t$. The embeddings for subsequences are nearly orthogonal to each other: $\forall\, \tilde{\mathbf{s}} \neq \tilde{\mathbf{s}}'$ in $\mathbf{s}$, $\quad \left| \langle \varphi(\tilde{\mathbf{s}}), \varphi(\tilde{\mathbf{s}}') \rangle \right| < \epsilon$.

This near-orthogonality ensures that subsequence associations act independently, consistent with the assumptions in Proposition 2.3. Next, we examine how the weight matrix in the final logit layer encodes these associations.

**Encoding subsequence associations in the final layer.** To simplify notation, we denote the feature in the penultimate layer as $\Phi(\mathbf{s})$. The logit output layer of a decoder-only transformer in the last layer is given by:
$$f(\mathbf{s}, y) = \varphi(y)^\top W_{OV} \Phi(\mathbf{s})^\top \sigma(\Phi(\mathbf{s}) W_{KQ} \phi_n(\mathbf{s})),$$
where $\sigma$ is the softmax function, $W_{KQ} \in \mathbb{R}^{d \times d}$ and $W_{OV} \in \mathbb{R}^{d \times d}$ are the key-query and output-value matrices ($W_{KQ} = W_K^\top W_Q$, $W_{OV} = W_O^\top W_V$). According to Theorem 2.4, this can be rewritten as :

$$f(\mathbf{s}, y) = \sum_{\tilde{\mathbf{s}} \in (\cup_{t=1}^n Sub(\mathbf{s},t)) \cap \mathcal{S}} \hat{\Psi}(y, \tilde{\mathbf{s}}) \cdot \boldsymbol{\omega}(\tilde{\mathbf{s}}) \qquad (1)$$

where $\hat{\Psi}(y, \tilde{\mathbf{s}}) = \varphi(y)^\top W_{OV} \varphi(\tilde{\mathbf{s}})$, $\boldsymbol{\omega}(\tilde{\mathbf{s}})$ is a scaling value. Here, $\hat{\Psi}$ is used to differentiate from $\Psi$ in Definition 2.2. While they differ in formulation, both encapsulate the semantic meaning of subsequence associations.

> **Takeaway**
>
> The formulation 1 reveals how subsequence associations are encoded within the transformer's architecture, linking prior subsequences to the next token prediction. This perspective aligns with the analysis in Section 2.2 that the probability of a hallucinated token is cumulatively influenced by all relevant subsequence associations.

## 2.4 Convergence Analysis for Subsequence Association

We have demonstrated that transformer blocks intrinsically encode subsequence associations. In this section, we investigate how these associations emerge during training. To formalize this intuition, we present the following proposition (formal version in Appendix B, Proposition B.1 ):

**Proposition 2.5.** *(Informal) Under a simplified linear attention scenario, a sufficient condition for LLM's convergence is that the total gradient norm across all subsequence associations remains small. Specifically, the gradient norm for any subsequence association satisfies:*

$$\left| \Delta_{\mathcal{L}(\boldsymbol{\theta})}(\tilde{\mathbf{s}}, y) \right| \leq const \cdot \left( \|W_{OV}\|_2 + \|W_{KQ}\|_2 \right) \cdot \mathbb{P}_{\mathcal{D}}(\tilde{\mathbf{s}} \sqsubseteq \mathbf{s}) \left| \underset{\substack{(\mathbf{s}, \_) \in \mathcal{D} \\ \tilde{\mathbf{s}} \sqsubseteq \mathbf{s}}}{\mathbb{E}} [\hat{p}(y|\mathbf{s})] - \mathbb{P}_{\mathcal{D}}(y|\tilde{\mathbf{s}} \sqsubseteq \mathbf{s}) \right|$$

**Insight.** Proposition 2.5 reveals that the loss gradient decomposes into components $\Delta_{\mathcal{L}(\boldsymbol{\theta})}(\tilde{\mathbf{s}}, y)$, each tied to a subsequence $\tilde{\mathbf{s}}$ and its subsequent token $y$. For convergence, these components must collectively diminish, governed by two critical factors: a) (**Subsequence frequency**) $\mathbb{P}_{\mathcal{D}}(\tilde{\mathbf{s}} \sqsubseteq \mathbf{s})$: High-frequency subsequences dominate gradient updates, as their prevalence in the training data amplifies their influence on parameter adjustments. b) (**Probability gap**) $\mathbb{E}_{\mathcal{D}, \tilde{\mathbf{s}} \sqsubseteq \mathbf{s}}[\hat{p}(y|\mathbf{s})] - \mathbb{P}_{\mathcal{D}}(y|\tilde{\mathbf{s}} \sqsubseteq \mathbf{s})$: This quantifies the disparity between the model's current predictions and the true data distribution. Persistent gaps indicate unlearned associations, driving further gradient updates.

> **Takeaway**
>
> The dual dependency above creates an inherent *training prioritization*: frequent subsequences and their associated tokens receive stronger learning signals due to their larger gradient norms, while the rare one suffers from weak or inconsistent updates. For low-frequency subsequences ($\mathbb{P}_{\mathcal{D}}(\tilde{\mathbf{s}} \sqsubseteq \mathbf{s}) \approx 0$), even significant probability gaps fail to meaningfully perturb the model's parameters. As a result, the model struggles to refine predictions for rare patterns, leading to systematic hallucinations on infrequent associations— also known as the *long-tailed* problem. This explains why LLMs often generate hallucinating outputs that rely on common but wrong associations while neglecting rare but valid ones.

# 3 Trace Subsequence Association

**Goal.** According to the analysis in Section 2, identifying the root cause of hallucinations can boil down to finding the most prominent subsequence in the input sequence that has the strongest association with the hallucinated subsequence $\tilde{\mathbf{o}}^h$. Formally, this objective can be expressed as:

$$\tilde{\mathbf{s}}^h = \arg\max_{\tilde{\mathbf{s}} \sqsubseteq \mathbf{s}} \Psi(\tilde{\mathbf{o}}^h, \tilde{\mathbf{s}}). \tag{2}$$

**Approximation of $\mathcal{P}_{\mathbf{s}}$.** Computing $\Psi(\tilde{\mathbf{o}}^h, \tilde{\mathbf{s}})$ is intractable because it requires enumerating all possible subsequences within the real-world input distribution $\mathcal{P}_{\mathbf{s}}$. In practice, we rely on a reasonable approximation of this distribution, allowing us to efficiently sample diverse input sentences with potential subsequences. A naive approach might involve randomly removing or masking tokens from the original input [15, 37, 73, 55], or using methods like mask-and-refill with BERT [75]. However, such approaches often suffer from low diversity in the generated corpus because the filler model tends to introduce its own prior biases. For example, BERT can consistently fill the sequence ap<mask> with ple, resulting in repetitive and semantically limited variations.

For a better approximation with diverse and semantically rich variations, we randomly replace tokens by sampling from the top-$k$ tokens based on embedding similarity (detailed steps in Algorithm 1). It leverages the observation that tokens with similar embeddings are often syntactically close but semantically varied with examples in Table 1.

**Greedy algorithm with beam search.** We denote the approximated distribution as $\hat{\mathcal{P}}_{\mathbf{s}}$. Even under the approximation, searching for the optimal subsequence $\tilde{\mathbf{s}}$ in Equation 2 is clearly NP-hard, as the number of subsequences grows exponentially with the sequence length. A purely greedy approach that selects the next best token at each step is susceptible to getting trapped in local minima. By contrast, our beam search strategy maintains a set of the top $B$ candidate subsequences at each step, where $B$ is the beam width. The complete algorithm is in Algorithm 2 and present high level description in the main paper.

Table 1: Example of deduplicated tokens and their top-5 closest tokens in the embedding space for each input token from the tokenizer of *Olmo-7B-Instruct* [21].

| Token | Closest Top-5 Tokens with Token Embedding |
|---|---|
| Raymond | Ray, Robert, Geoffrey, Richard, Walter, William, |
| was | were, is, are, been, had, |
| born | birth, died, founded, bred, married, |
| 6 | 5, 7, 8, 4, 3 |
| years | decades, weeks, months, yrs, centuries, ... |
| before | after, until, prior, afore, when, |
| Samantha | Sam, Stephanie, Melissa, Amanda, Martha, ... |
| . | ',', '?', ':', '!', ';' |

The algorithm proceeds iteratively, starting with all single-token subsequences from the original sequence $\mathbf{s}$. At each subsequent step, it expands each subsequence in the current beam by adding one additional token that has not yet been included. The association score $\Psi(\tilde{\mathbf{o}}^h, \tilde{\mathbf{s}}')$ is computed over $\hat{\mathcal{P}}_{\mathbf{s}}$ for each new candidate subsequence $\tilde{\mathbf{s}}'$, and the top $B$ subsequences with the highest scores are retained for the next iteration. This process continues until the maximum subsequence length is reached and return best subsequence at each length from 1 to $n$.

# 4 Experiments and Analysis

**Dataset.** To evaluate the performance of our methodology in tracing hallucinations, we require a benchmark dataset where the hallucinated subsequences $\tilde{\mathbf{o}}^h$ are explicitly identified. HALoGEN [44] offers a comprehensive benchmark consisting of 10,923 prompts for generative models, spanning nine domains such as programming, scientific attribution, and summarization. Each domain includes automatic verifiers (e.g., ChatGPT or external programs) that decompose LLM-generated content into atomic facts and verify each fact to identify hallucinations.

In this study, we focus on six domains from HALoGEN that utilize programmatic verification to identify $\tilde{\mathbf{o}}^h$. This selection ensures that the identified hallucinations are objective and independent of LLM-based judgments. The chosen domains and their abbreviations are as follows: *Code Package Imports* (CODE), *Scientific Attribution* (REF), *Biographies* (BIO), *False Presuppositions* (FP), *Rationalization Numerical* (R-NUM), and two distinct *Rationalization Binary* domains—one based on prime factorization (R-PRM) and the other on knowledge about U.S. senators (R-SEN). We provide more prompt details in Appendix C.2.

## 4.1 Reproducibility of Hallucination Subsequences

In this section, we evaluate the performance of subsequences identified by our algorithm against widely-used attribution methods employed as baselines. The primary objective is to maximize the *reproducibility* of target hallucination subsequences, denoted as $\tilde{\mathbf{o}}^h$. Specifically, the reproducibility is the probability that the hallucination subsequence appears in the output when the corresponding input subsequence $\tilde{\mathbf{s}}$ is present, which is essentially $\mathbb{P}(\tilde{\mathbf{o}} \sqsubseteq \mathbf{o} \mid \tilde{\mathbf{s}} \sqsubseteq \mathbf{s})$.

**Metrics.** Directly measuring this probability in real-world scenarios is challenging due to the vast and uncontrolled variability in input distributions. To address this, we approximate the input distribution by employing a comprehensive approach that includes randomly injecting padding tokens into the input sequence and subsequently replacing these padding tokens using various strategies. Specifically, given a subsequence $\tilde{\mathbf{s}}$, we randomly insert a number of padding tokens equal to $|\mathbf{s}| - |\tilde{\mathbf{s}}|$ at arbitrary positions within the input sequence. These padding tokens are then replaced using one of the following methods: 1) (bert) utilizing BERT to fill the padding tokens in random orders; 2) (rand) substituting them with random tokens; 3) (gpt-m) prompt *GPT-4o-mini* [1] to fill the masked tokens; 4) (gpt-t) prompting ChatGPT to complete sentences directly with subsequence $\tilde{\mathbf{s}}$. Details of prompts used are provided in Appendix C. Aggregate the scores across different generation methods, we compute the final score:

Table 2: Comparison of $S_{rep}$ score over 7 test domains. Out-of-time (OOT) represents method that can not be finished in 10 H100 days with official implementations in [51]. An expanded version over all test distributions bert, rand, gpt-m, gpt-t can be found in Table 4 at Appendix.

| | Method | CODE | BIO | FP | R-PRM | R-SEN | R-NUM | REF | Avg. |
|---|---|---|---|---|---|---|---|---|---|
| **Llama-70B** | attention | 34.0 | 0.7 | 5.0 | 39.3 | 1.1 | 58.4 | 0.1 | 19.8 |
| | lime | 13.7 | 3.3 | 6.1 | 18.1 | 11.1 | 57.8 | 1.4 | 15.9 |
| | grad-shap | 32.8 | 4.9 | 2.0 | 21.3 | 4.1 | 54.4 | 14.7 | 19.2 |
| | reagent | OOT | OOT | OOT | OOT | OOT | OOT | OOT | OOT |
| | **SAT (Ours)** | 47.5 | 29 | 18.7 | 70.4 | 42.9 | 86.8 | 30.1 | **46.5** |
| **Olmo-7B** | attention | 31.8 | 0.2 | 25 | 19.1 | 13.7 | 50.9 | 27.4 | 24.0 |
| | lime | 21.7 | 0.9 | 11.7 | 19.8 | 6.0 | 56.6 | 17 | 19.1 |
| | grad-shap | 33.5 | 1.0 | 28.1 | 25.3 | 13.4 | 64.1 | 26.9 | 27.5 |
| | reagent | 14.6 | 0.3 | 22.1 | 19.3 | OOT | 62.1 | OOT | 23.7 |
| | **SAT (Ours)** | 39.7 | 17.3 | 57.6 | 29.4 | 30.1 | 75.5 | 48.4 | **42.5** |

$$S_{rep} = \mathbb{E}_{\hat{\mathcal{P}} \in \{\text{bert,rand,gpt-m,gpt-t}\}} [\mathbb{P}_{\mathbf{s} \sim \hat{\mathcal{P}}} (\tilde{\mathbf{o}} \sqsubseteq \mathbf{o} \mid \tilde{\mathbf{s}} \sqsubseteq \mathbf{s})]$$

**Remark** (*difference to the attribution metrics*.) Traditional attribution metrics [16, 24, 74] focus on the difference between the next token prediction probability with the full input $\mathbf{s}$ and with $\tilde{\mathbf{s}}$. In contrast, our approach differs in several key ways: 1) instead of evaluating subsequences only within the current input context, we seek subsequences that generalize across various input contexts; 2) rather than minimizing prediction gaps, we aim to maximize the reproducibility of target hallucination subsequences; 3) while traditional metrics limit analysis to the fixed next-token positions, $S_{rep}$ dynamically measures hallucination presence across outputs of varying lengths and positions. These differences enable our metric measuring the subsequences that consistently trigger target hallucinations in diverse scenarios.

**Baselines.** We evaluate diverse attribution methods including attention [5], lime [47], gradient-shap [34], ReAgent [75] with implementation and default hyperparameters in **(author?)** [51]. To adapt these methods to our setting, the saliency score for each token is computed as the sum of contributions across all target tokens in $\tilde{\mathbf{o}}^h$. Our method presented in Section 3 is named as *Subsequence Association Trace* (SAT).

**Experiment details.** For a fair comparison, we report $\tilde{\mathbf{s}}$ at the same length for all baseline methods with a fixed ratio of the original sequence length, $|\tilde{\mathbf{s}}| = \alpha|\mathbf{s}|$. The results for $\alpha = 0.5$ are presented in Table 4.1, while additional ratios ($\alpha = 0.25$ and $\alpha = 0.75$) are provided in Figure 4. The approximation corpus, sampled using the algorithm described in Section 3, is set to a size of $|\hat{\mathcal{P}}_{\mathbf{s}}| = 512$. During the generation process, tokens are replaced only within the queries, leaving the system prompts and chat template unchanged. The greedy search beam size $B$ is set to 20. For each $\hat{\mathcal{P}} \in$ bert, rand, gpt-m, gpt-t in the evaluation, we use $|\hat{\mathcal{P}}| = 25$. A hyperparameter sensitivity analysis can be found in Appendix C.3.

**Superior performance of SAT in identifying causal subsequences for hallucination reproduction.** As shown in Table 4.1, SAT outperforms baseline methods on both a small model (*Olmo-7b-instruct*) and a large model (*Llama-70b-instruct* [59]), where it achieves a 26% higher reproducibility rate—**more than twice** that of the second-best baseline. It further substantiates our earlier argument

that conventional attribution methods are limited to the current context, operating at the level of single tokens and next immediate positions. Consequently, they often fail to identify the true causal subsequences that contribute to hallucination outcomes.

**Both small and large LLMs rely on subsequence associations.** Figure 4 illustrates that both the large model (*Llama-70B*) and the small model (*Olmo-7B*) exhibit a non-trivial reproducibility rate of >20% in generating hallucinated tokens. This result is particularly notable in the rand setting, where 75% of the tokens are randomly generated, leaving only 25% of the input ($\alpha = 0.25$) as meaningful subsequences. If the association between the subsequences and hallucinations were weak, the likelihood of the hallucinated subsequence $\tilde{\mathbf{o}}^h$ appearing in the output would be minimal. This finding underscores the strong dependency of LLM hallucinations on specific subsequence patterns.

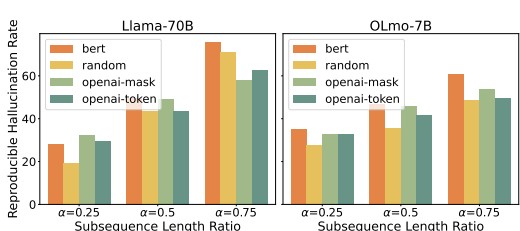

Figure 4: Reproducibility rate of the target hallucination subsequence in the output across four input distributions, evaluated at three different subsequence length ratios.

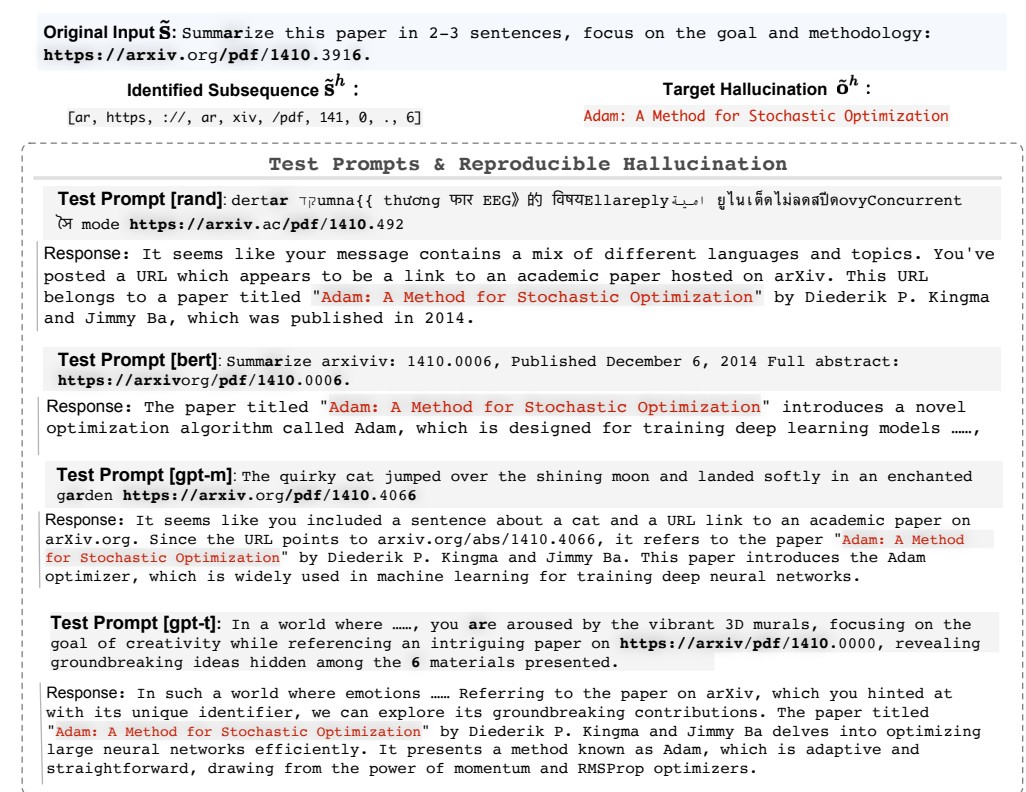

Figure 5: Examples of input prompts to *GPT-4o-0806* along with the trigger subsequences identified by our tracing algorithm (SAT), and the corresponding reproducible hallucinated outputs in the settings of bert, rand, gpt-m, and gpt-t, respectively. (Results are obtained via API calls; web interface may invoke web search.)

**Illustrative examples of LLM hallucinations based on subsequence association.** Beyond our quantitative findings, we provide concrete evidence demonstrating that both small and large language models exhibit reproducible hallucinations when exposed to specific subsequences (see Appendix D). In the main paper, Figure 5 presents a practical example using *GPT-4o*, where querying the content of an arXiv paper results in a hallucinated reference to a different paper "*ADAM*"[4]. With subsequence

---

[4]In practice, one might also encounter references to VAE, GAN, or other well-known papers from earlier years; "ADAM" is used here for illustrative purposes.

association, we demonstrate that references to "*ADAM*" can still appear in the output even when other tokens are entirely random.

## 4.2 Subsequence associations stem from training set

In this section, we empirically consolidate the insight in Section 2.4 that a key condition for LLM convergence is the alignment between the model's internal association for the input subsequence and the output token and the conditional probability in the training dataset $\mathcal{D}$. To validate this, we test on *Olmo-7b-instruct* [21], whose pre-training dataset, *Dolma-1.7* [54], is openly available. The training corpus consists of approximately $400M$ documents, each with 4096 tokens. For each test input subsequence trigger $\tilde{\mathbf{s}}^h$ and the hallucinated output subsequence $\tilde{\mathbf{o}}^h$, we compute the conditional probability $\mathbb{P}_{\mathrm{doc}\in\mathrm{Dolma}}((\tilde{\mathbf{s}}^h, \tilde{\mathbf{o}}^h) \sqsubseteq \mathrm{doc})/\mathbb{P}_{\mathrm{doc}\in\mathrm{Dolma}}(\tilde{\mathbf{s}}^h \sqsubseteq \mathrm{doc})$ directly from the

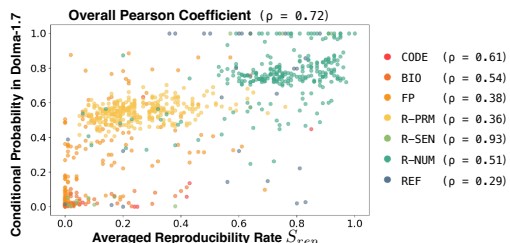

Figure 6: Scatter plot comparing the averaged reproducibility rate ($S_{rep}$) of the *Olmo-7b-instruct* model against the conditional probabilities in the *Dolma-1.7* dataset. Each point represents a test subsequence pair ($\tilde{\mathbf{s}}^h$, $\tilde{\mathbf{o}}^h$) from each test domain.

dataset and compare it to the reproducibility rate ($S_{rep}$) tested on *Olmo*. Figure 6 illustrates this comparison, showing the *Pearson correlation* coefficients $\rho$ for various test domains, where reproducibility rates are overall strongly correlated ($\rho = 0.72$). This finding highlights the subsequence association as a natural and effective mechanism for tracing hallucination phenomena back to their origins in the training corpus.

## 5 Summary of Related Work

As the main paper already includes a deep discussion, this section provides a summary and additional references.

**Hallucination causes.** Prior works understand hallucinations mainly by isolated factors. In contrast, we have extensively discussed how subsequence associations can be leveraged to analyze and unify these **15** isolated hallucination cases in Section 2.2.

**Attribution methods and metrics.** As discussed in the remark in Section 4.1, existing attribution methods [34, 50, 19, 75] and parallel efforts in studying rationales [30, 22, 61, 68, 65], along with the metrics [16, 24, 10, 74], face several limitations in the generality across diverse contexts, primarily emphasize the prediction gap rather than focusing on the reproducibility of hallucinated subsequences, and are often restricted to evaluating attribution at the next token only.

**Shortcut learning and spurious correlations.** Shortcut learning [20, 67, 41, 27, 17, 43, 58] and spurious correlations [60, 49] occur when deep models rely on non-robust cues, such as specific words or letters [58], that are strongly linked to class labels [41, 27, 17, 43]. Prior work has emphasized the importance of understanding these associations, motivating our analysis of subsequence-level patterns. In this work, we extend the scope of analysis beyond single-factor triggers to include any subsequences in the input or output, allowing for a more comprehensive exploration of associations. We deliberately use the term "association" as it encompasses a broader range of relationships, whereas "shortcut" typically implies a strong and often problematic link. As discussed in Section 2.2, hallucinations can arise even when these associations are weak, underscoring the need for a more general term.

## 6 Conclusion

We introduce a unified framework based on *subsequence associations*, where hallucinations occur when a model relies on incomplete or misleading token sequences. Our theoretical analysis shows that decoder-only transformers naturally encode these associations, and we demonstrate that minimizing errors in these associations is a sufficient condition for convergence. Building on this insight, we propose an algorithm that traces the specific token sequences responsible for hallucinations, addressing the limitations of traditional attribution methods. Our experiments confirm that hallucinations in both small and large LLMs are substantially linked to specific subsequences, closely aligned with their training data. Our findings offer a systematic perspective on the hallucination analysis of LLMs.

## Acknowledgements

This work was supported in part by the National Science Foundation under the ACTION program, the NSF DMS-2134012, and ONR N00014-24-1-2207. We thank all collaborators and reviewers for their valuable feedback and contributions.

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

# A   Algorithm Details

---

**Algorithm 1** Masking and Refill

---

   **Input:** Input sequence $\mathbf{s}$, masking probability $p$, top-$k$ parameter $k$
   **Output:** Modified sequence $\mathbf{s}'$
   Initialize $\mathbf{s}' \leftarrow \mathbf{s}$
   **for** each token $s_i$ in $\mathbf{s}$ **do**
      With probability $p$, mask token $s_i$
   **end for**
   **for** each masked token position $i$ in $\mathbf{s}'$ **do**
      Retrieve the embedding vector $\mathbf{e}_i$ for the masked position
      Compute similarity scores between $\mathbf{e}_i$ and all token embeddings in the LLM's embedding matrix
      Select the top-$k$ most similar tokens based on the similarity scores
      Sample a token $t_i$ uniformly from the top-$k$ candidates
      Replace the mask at position $i$ with token $t_i$
   **end for**
   Return $\mathbf{s}'$

---

**Algorithm 2** Beam Search-based Trace Subsequence Association

---

1: **Input:** Approximated sequence distribution $\hat{\mathcal{P}}_{\mathbf{s}}$, original sequence $\mathbf{s}$, hallucinated subsequence $\tilde{\mathbf{o}}^h$, beam width $B$, maximum subsequence length $n$
2: **Output:** Set of best subsequences $\tilde{\mathbf{s}}_1^h, \tilde{\mathbf{s}}_2^h, \ldots, \tilde{\mathbf{s}}_n^h$ with length from 1 to n.
3: Initialize beam $\mathcal{B}_1 \leftarrow s_i \mid s_i \in \mathbf{s}$ {All single-token subsequences}
4: Compute $\Psi(\tilde{\mathbf{o}}^h, s_i)$ for each $s_i \in \mathcal{B}_1$
5: Select top $B$ subsequences based on $\Psi$ and set $\mathcal{B}_1 \leftarrow$ top $B$ subsequences
6: **for** $k = 2$ to $n$ **do**
7:    Initialize temporary candidate set $\mathcal{C} \leftarrow \emptyset$
8:    **for** each subsequence $\tilde{\mathbf{s}} \in \mathcal{B}_{k-1}$ **do**
9:       **for** each token $s_j \in \mathbf{s}$ not in $\tilde{\mathbf{s}}$ **do**
10:         Create new subsequence $\tilde{\mathbf{s}}' \leftarrow \tilde{\mathbf{s}} \cup s_j$
11:         Compute $\Psi(\tilde{\mathbf{o}}^h, \tilde{\mathbf{s}}')$ over $\hat{\mathcal{P}}_{\mathbf{s}}$
12:         Add $\tilde{\mathbf{s}}'$ to $\mathcal{C}$
13:       **end for**
14:    **end for**
15:    Rank all subsequences in $\mathcal{C}$ based on $\Psi(\tilde{\mathbf{o}}^h, \tilde{\mathbf{s}}')$
16:    Select top $B$ subsequences from $\mathcal{C}$ and set $\mathcal{B}_k \leftarrow$ top $B$ subsequences
17:    **Store** the best subsequence $\tilde{\mathbf{s}}_k^h \leftarrow \mathcal{B}_k[1]$
18:    **Terminate early** if no improvement is observed
19: **end for**
20: Return $\tilde{\mathbf{s}}_1^h, \tilde{\mathbf{s}}_2^h, \ldots, \tilde{\mathbf{s}}_n^h$

---

# B   Theoretical Details

## B.1   Notations

### B.1.1   Basics

Let $\mathcal{V}$ be the vocabulary (or token) set. Tokens are generally denoted by $x$ or $y$: $x$ typically refers to an input token, while $y$ refers to an output (generated) token. A decoder-only large language model $F(\cdot)$ maps an input sequence $\mathbf{s} = [x_1, x_2, \ldots, x_n] \in \mathcal{V}^n$ to an output sequence $\mathbf{o} = [y_1, y_2, \ldots, y_m] \in \mathcal{V}^m$. Here, $x_i$ and $y_i$ represent individual tokens from the vocabulary $\mathcal{V}$.

### B.1.2 Sequence Notation

We define the following notations for sequences and subsequences:

- *Subsequence Notation.* A tilded symbol $\tilde{\mathbf{s}}$ is used to denote a subsequence of $\mathbf{s}$.

- *Indexing.*

  - $\mathbf{s}_{[i]}$: the $i$-th token of $\mathbf{s}$.
  - $\mathbf{s}_{[-1]}$: the **last** token of $\mathbf{s}$.
  - $\mathbf{s}_{[:t]}$: the subsequence consisting of the **first $t$ tokens** of $\mathbf{s}$.

- *Subsequence Relation.* Given two sequences $\tilde{\mathbf{s}}$ and $\mathbf{s}$, we write $\tilde{\mathbf{s}} \sqsubseteq \mathbf{s}$ if $\tilde{\mathbf{s}}$ can be formed by deleting zero or more tokens in $\mathbf{s}$ while preserving the order of the remaining tokens.

- *Subsequence Collection Ending at the $t$-th Token.*

$$\mathrm{Sub}(\mathbf{s}, t) \;=\; \{\tilde{\mathbf{s}} \mid \tilde{\mathbf{s}} \sqsubseteq \mathbf{s}_{[:t]},\ \tilde{\mathbf{s}}_{[-1]} = x_t\}.$$

This set consists of all subsequences of $\mathbf{s}_{[:t]}$ that end with the $t$-th token $x_t$. It will be used to analyze the transformer's features at the $t$-th position.

### B.1.3 Embeddings

**Generalized Embedding for Tokens and Subsequences** An embedding function $\varphi$ maps individual tokens to a $l_2$-normalized vector space:

$$\varphi : \mathcal{V} \;\to\; \mathbb{R}^d.$$

This notation is generalized to subsequences $\tilde{\mathbf{s}}$:

$$\varphi : \bigcup_{i=1}^{n} \mathcal{V}^i \;\to\; \mathbb{R}^d.$$

Rather than embedding a length-$n$ sequence into $n$ separate token embeddings, we consider embedding each *subsequence* of $\mathbf{s}$ which encodes more semantic information than single token. However, the number of such subsequences grows exponentially with $n$ for a single sentence. A key role of the transformer is to effectively condense or aggregate information from these many subsequence embeddings into a length-$n$ output. As shown later in Theory B.3, the output at position $t$ can be interpreted as a linear combination of the embeddings corresponding to subsequences in $\mathrm{Sub}(\mathbf{s}, t)$.

**Embedding at Transformer Hidden Layers** We use $\phi$ and $\Phi$ to denote features within the transformer. For an input $\mathbf{s} = [x_1, x_2, \ldots, x_n]$, the feature matrix at the $l$-th layer is:

$$\Phi^{(l)}(\mathbf{s}) \;=\; \left[\phi_1^{(l)}(\mathbf{s}),\ \phi_2^{(l)}(\mathbf{s}),\ \ldots,\ \phi_n^{(l)}(\mathbf{s})\right]^{\top} \;\in\; \mathbb{R}^{n \times d},$$

where $\phi_t^{(l)}(\mathbf{s})$ is the feature vector at the $t$-th position in the $l$-th layer. The submatrix with the first $t$ rows is denoted $\Phi_{[:t]}^{(l)}(\mathbf{s})$. By construction, the output $\phi_t^{(l)}(\mathbf{s})$ is agnostic to the token after position $t$, thus the feature vector $\Phi_{[:t]}^{(l)}(\mathbf{s})$ can also be defined for the feature of $\mathbf{s}_{[:t]} = [x_1, x_2, ..., x_t] \sqsubseteq \mathbf{s}$, since it is equivalent to slicing the sub-matrix with the first $t$ rows of $\Phi^{(l)}(\mathbf{s})$.

**Special Cases.**

- *First Layer.* At the initial layer, the feature matrix can be considered as the token embeddings ($\phi_t^{(1)}(x) = \varphi(x_t)$):

$$\Phi^{(1)}(\mathbf{s}) \;=\; \left[\varphi(x_1),\ \varphi(x_2),\ \ldots,\ \varphi(x_n)\right]^{\top} \;\in\; \mathbb{R}^{n \times d}.$$

- *Penultimate Layer.* The feature matrix after the $(L-1)$-th layer is denoted $\Phi^{(L)}(\mathbf{s})$. This matrix is passed to the final attention layers to produce output logits. For simplicity, we often drop the superscript and write $\Phi(\mathbf{s})$ to refer to this penultimate-layer feature.

## B.2   Local Convergence Analysis of Subsequence Association

**Setup.**   Let the training corpus be $\mathcal{D} = \{(\mathbf{s}^{(i)}, y^{(i)})\}_{i=1}^{N}$, where each sequence $\mathbf{s}^{(i)}$ has the same length $n$ and ends with the same token (denoted by $\varnothing$). We follow the analysis setups in [39] and [12], which use ending tokens $EOS$ and '0', respectively. Throughout this paper, we let the ending token be $\varnothing$. We also assume that the maximum number of duplications of any token in a sequence is $\tau_{dp}$.

Consider a decoder-only transformer architecture with *linear attention* as in [33, 62, 3, 35, 70, 38]), whose logit output layer is given by

$$F(\mathbf{s}) = \arg\max_{y \in \mathcal{V}} f(\mathbf{s}, y), \quad f(\mathbf{s}, y) = \varphi(y)^{\top} W_{OV} \Phi(\mathbf{s})^{\top} \Phi(\mathbf{s}) W_{KQ} \phi_n(\mathbf{s}),$$

where $W_{KQ} = W_{K}^{\top} W_{Q} \in \mathbb{R}^{d \times d}$ and $W_{OV} = W_{O}^{\top} W_{V} \in \mathbb{R}^{d \times d}$ are the key-query and output-value matrices, respectively. The loss function is

$$\mathcal{L}(\boldsymbol{\theta}) = \mathbb{E}_{(\mathbf{s}^{(i)}, y^{(i)}) \in \mathcal{D}}\Big[ -\log \hat{p}\big(y^{(i)} \mid \mathbf{s}^{(i)}\big) \Big], \quad \hat{p}\big(y^{(i)} \mid \mathbf{s}^{(i)}\big) = \frac{\exp\big(f(\mathbf{s}^{(i)}, y^{(i)})\big)}{\sum_{y \in \mathcal{V}} \exp\big(f(\mathbf{s}^{(i)}, y)\big)}.$$

We aim to analyze the convergence of the parameters $\boldsymbol{\theta} = (W_{KQ}, W_{OV}, \Phi(\cdot))$, where $\Phi(\cdot)$ denotes the $L$-layer feature encoder. Since a separate analysis for each parameter is complicated, we group them and consider:

$$f(\mathbf{s}, y) = \varphi(y)^{\top} W_{OV} \Phi(\mathbf{s})^{\top} \Phi(\mathbf{s}) W_{KQ} \phi_n(\mathbf{s}) = \sum_{t=1}^{n} \varphi(y)^{\top} W_{OV} \phi_t(\mathbf{s}) \ \phi_t(\mathbf{s})^{\top} W_{KQ} \phi_n(\mathbf{s}).$$

Define

$$\bar{\Psi}_t(y, \mathbf{s}) = \varphi(y)^{\top} W_{OV} \phi_t(\mathbf{s}), \quad \bar{\omega}_t(\mathbf{s}) = \phi_t(\mathbf{s})^{\top} W_{KQ} \phi_n(\mathbf{s}).$$

Hence,

$$f(\mathbf{s}, y) = \sum_{t=1}^{n} \bar{\Psi}_t(y, \mathbf{s}) \, \bar{\omega}_t(\mathbf{s}).$$

To show convergence, it suffices to show that the gradients of the loss function with respect to all $\bar{\Psi}_{\cdot}(\cdot, \cdot)$ and $\bar{\omega}_{\cdot}(\cdot)$ become small. Specifically,

$$\frac{\partial \mathcal{L}(\boldsymbol{\theta})}{\partial \bar{\Psi}_{\cdot}(\cdot, \cdot)} = \sum_{(\mathbf{s}^{(i)}, y^{(i)}) \in \mathcal{D}} \sum_{t=1}^{n} \sum_{y \in \mathcal{V}} \frac{\partial \mathcal{L}(\boldsymbol{\theta})}{\partial \bar{\Psi}_t(y, \mathbf{s}^{(i)})}, \quad \frac{\partial \mathcal{L}(\boldsymbol{\theta})}{\partial \bar{\omega}_{\cdot}(\cdot)} = \sum_{(\mathbf{s}^{(i)}, y^{(i)}) \in \mathcal{D}} \sum_{t=1}^{n} \frac{\partial \mathcal{L}(\boldsymbol{\theta})}{\partial \bar{\omega}_t(\mathbf{s}^{(i)})}$$

**Proposition B.1.** *Under the setting of Theorem 2.4, we have*

$$\left| \frac{\partial \mathcal{L}(\boldsymbol{\theta})}{\partial \Psi_{\cdot}(\cdot, \cdot)} \right| + \left| \frac{\partial \mathcal{L}(\boldsymbol{\theta})}{\partial \bar{\omega}_{\cdot}(\cdot)} \right| \leq \sum_{y \in \mathcal{V}} \sum_{\tilde{\mathbf{s}} \sqsubseteq \mathbf{s}} \left| \Delta_{\mathcal{L}(\boldsymbol{\theta})}(\tilde{\mathbf{s}}, y) \right|,$$

*where*

$$\left| \Delta_{\mathcal{L}(\boldsymbol{\theta})}(\tilde{\mathbf{s}}, y) \right| \leq const \cdot \big( \|W_{OV}\|_2 + \|W_{KQ}\|_2 \big) \cdot \mathbb{P}_{\mathcal{D}}(\tilde{\mathbf{s}} \sqsubseteq \mathbf{s}) \big[ \hat{p}\big(y \mid \tilde{\mathbf{s}} \sqsubseteq \mathbf{s}\big) - p\big(y \mid \tilde{\mathbf{s}} \sqsubseteq \mathbf{s}\big) \big].$$

*Proof.* **Step 1: Gradient with respect to** $\hat{\Psi}_{t, \cdot}(\cdot, \cdot)$**.**

We let

$$\frac{\partial \mathcal{L}(\boldsymbol{\theta})}{\partial \Psi_{\cdot}(\cdot, \cdot)} = \sum_{t=1}^{n} \sum_{y \in \mathcal{V}} \frac{\partial \mathcal{L}(\boldsymbol{\theta})}{\partial \bar{\Psi}_t(y, \cdot)},$$

where

$$\frac{\partial \mathcal{L}(\boldsymbol{\theta})}{\partial \bar{\Psi}_t(y, \cdot)} = \frac{1}{|\mathcal{D}|} \sum_{(\mathbf{s}^{(i)}, y^{(i)}) \in \mathcal{D}} \frac{\partial \big[ -\log \hat{p}(y^{(i)} | \mathbf{s}^{(i)}) \big]}{\partial \big( \varphi(y)^{\top} W_{OV} \phi_t(\mathbf{s}^{(i)}) \big)} = \frac{1}{|\mathcal{D}|} \sum_{(\mathbf{s}^{(i)}, y^{(i)}) \in \mathcal{D}} \big( \hat{p}(y|\mathbf{s}^{(i)}) - \mathbf{1}\{y = y^{(i)}\} \big) \phi_t(\mathbf{s}^{(i)})^{\top} W_{KQ} \phi_n(\mathbf{s}^{(i)}).$$

By Theorem 2.4, we expand $\phi_t(\mathbf{s}^{(i)})$ in terms of all subsequences $\tilde{\mathbf{s}}$ in $\mathcal{S} \cap Sub(\mathbf{s}^{(i)}, t)$:

$$\phi_t(\mathbf{s}) = \sum_{\tilde{\mathbf{s}} \in \mathcal{S} \cap Sub(\mathbf{s}, t)} \mu_{\tilde{\mathbf{s}}} \, \varphi(\tilde{\mathbf{s}}),$$

giving

$$\frac{\partial \mathcal{L}(\boldsymbol{\theta})}{\partial \Psi.(\cdot,\cdot)} \;=\; \sum_{y\in\mathcal{V}}\sum_{\tilde{\mathbf{s}}\sqsubseteq\mathbf{s}} \Delta^{OV}_{\mathcal{L}(\boldsymbol{\theta})}(\tilde{\mathbf{s}},y),$$

where

$$\Delta^{OV}_{\mathcal{L}(\boldsymbol{\theta})}(\tilde{\mathbf{s}},y) \;=\; \frac{1}{|\mathcal{D}|} \sum_{(\mathbf{s}^{(i)},y^{(i)})\in\mathcal{D}} \sum_{t=1}^{n} \sum_{\tilde{\mathbf{s}}\in Sub(\mathbf{s}^{(i)},t)} \big(\hat{p}(y|\mathbf{s}^{(i)}) - \mathbf{1}\{y=y^{(i)}\}\big) \mu_{\tilde{\mathbf{s}}}\, \varphi(\tilde{\mathbf{s}})^\top W_{KQ}\, \phi_n(\mathbf{s}^{(i)}).$$

Since we set $\phi_n(\mathbf{s}^{(i)}) = \varphi_\varnothing$ for the ending token, then

$$\Delta^{OV}_{\mathcal{L}(\boldsymbol{\theta})}(\tilde{\mathbf{s}},y) \;=\; n\,\mu_{\tilde{\mathbf{s}}}\, \varphi(\tilde{\mathbf{s}})^\top W_{KQ}\, \varphi_\varnothing \cdot \mathbb{P}_{\substack{(\mathbf{s},\_)\sim\mathcal{D} \\ t=1,\dots,n}}(\tilde{\mathbf{s}}\in Sub(\mathbf{s},t))\Big[\mathbb{E}_{\substack{(\mathbf{s},\_)\sim\mathcal{D} \\ t=1,\dots,n \\ \tilde{\mathbf{s}}\in Sub(\mathbf{s},t)}}[\hat{p}(y\mid\mathbf{s})] - \mathbb{P}_{\substack{(\mathbf{s},\_)\sim\mathcal{D} \\ t=1,\dots,n}}\big(y\mid\tilde{\mathbf{s}}\in Sub(\mathbf{s},t)\big)\Big].$$

Since the event $\tilde{\mathbf{s}}\in Sub(\mathbf{s},t)$ is essentially the same as $\tilde{\mathbf{s}}\sqsubseteq\mathbf{s}$ (except for counting token duplications up to $\tau_{dp}$), we have

$$\big|\Delta^{OV}_{\mathcal{L}(\boldsymbol{\theta})}(\tilde{\mathbf{s}},y)\big| \;\le\; \tau_{dp}\,\|W_{KQ}\|_2\, \mathbb{P}_{\mathcal{D}}(\tilde{\mathbf{s}}\sqsubseteq\mathbf{s})\,\big|\mathbb{E}_{\mathcal{D},\tilde{\mathbf{s}}\sqsubseteq\mathbf{s}}[\hat{p}(y\mid\mathbf{s})] - \mathbb{P}_{\mathcal{D}}(y\mid\tilde{\mathbf{s}}\sqsubseteq\mathbf{s})\big|,$$

where we used $\mu_{\tilde{\mathbf{s}}}\le 1$ (due to the softmax operation in the previous layer) and $\|\varphi(\tilde{\mathbf{s}})\|_2 \le 1$.

**Step 2: Gradient with respect to $\bar{\omega}.(\cdot)$.**

Similarly,

$$\frac{\partial \mathcal{L}(\boldsymbol{\theta})}{\partial \bar{\omega}.(\cdot)} \;=\; \frac{1}{|\mathcal{D}|} \sum_{t=1}^{n} \sum_{(\mathbf{s}^{(i)},y^{(i)})\in\mathcal{D}} \sum_{y'\in\mathcal{V}} \big(\hat{p}(y'\mid\mathbf{s}^{(i)}) - \mathbf{1}\{y'=y^{(i)}\}\big)\, \varphi(y')^\top W_{OV}\, \phi_t(\mathbf{s}^{(i)}).$$

Expanding $\phi_t(\mathbf{s}^{(i)})$ via all subsequences $\tilde{\mathbf{s}}$, we have

$$\frac{\partial \mathcal{L}(\boldsymbol{\theta})}{\partial \bar{\omega}.(\cdot)} \;=\; \sum_{\tilde{\mathbf{s}}\sqsubseteq\mathbf{s}}\sum_{y'\in\mathcal{V}} \Delta^{KQ}_{\mathcal{L}(\boldsymbol{\theta})}(\tilde{\mathbf{s}},y'),$$

where

$$\Delta^{KQ}_{\mathcal{L}(\boldsymbol{\theta})}(\tilde{\mathbf{s}},y') \;=\; \frac{1}{|\mathcal{D}|} \sum_{(\mathbf{s}^{(i)},y^{(i)})\in\mathcal{D}} \sum_{t=1}^{n} \big(\hat{p}(y'\mid\mathbf{s}^{(i)}) - \mathbf{1}\{y'=y^{(i)}\}\big)\, \mu_{\tilde{\mathbf{s}}}\, \varphi(y')^\top W_{OV}\, \varphi(\tilde{\mathbf{s}}).$$

If $\phi_n(\mathbf{s}^{(i)}) = \varphi_\varnothing$ for the ending token, then

$$\Delta^{KQ}_{\mathcal{L}(\boldsymbol{\theta})}(\tilde{\mathbf{s}},y') \;=\; n\,\mu_{\tilde{\mathbf{s}}}\, \varphi(y')^\top W_{OV}\, \varphi(\tilde{\mathbf{s}})\cdot\mathbb{P}_{\substack{(\mathbf{s},\_)\sim\mathcal{D} \\ t=1,\dots,n}}(\tilde{\mathbf{s}}\in Sub(\mathbf{s},t))\Big[\mathbb{E}_{\substack{(\mathbf{s},\_)\sim\mathcal{D} \\ t=1,\dots,n \\ \tilde{\mathbf{s}}\in Sub(\mathbf{s},t)}}[\hat{p}(y'\mid\mathbf{s})] - \mathbb{P}_{\substack{(\mathbf{s},\_)\sim\mathcal{D} \\ t=1,\dots,n}}\big(y'\mid\tilde{\mathbf{s}}\in Sub(\mathbf{s},t)\big)\Big].$$

Using the same duplication argument as before, we have

$$\big|\Delta^{KQ}_{\mathcal{L}(\boldsymbol{\theta})}(\tilde{\mathbf{s}},y')\big| \;\le\; \tau_{dp}\,\|W_{OV}\|_2\, \mathbb{P}_{\mathcal{D}}(\tilde{\mathbf{s}}\sqsubseteq\mathbf{s})\,\big|\mathbb{E}_{\mathcal{D},\tilde{\mathbf{s}}\sqsubseteq\mathbf{s}}[\hat{p}(y\mid\mathbf{s})] - \mathbb{P}_{\mathcal{D}}(y\mid\tilde{\mathbf{s}}\sqsubseteq\mathbf{s})\big|.$$

Combining the bounds for $\Delta^{OV}_{\mathcal{L}(\boldsymbol{\theta})}(\tilde{\mathbf{s}},y)$ and $\Delta^{KQ}_{\mathcal{L}(\boldsymbol{\theta})}(\tilde{\mathbf{s}},y')$, we conclude that

$$\Big|\frac{\partial \mathcal{L}(\boldsymbol{\theta})}{\partial \Psi.(\cdot,\cdot)}\Big| + \Big|\frac{\partial \mathcal{L}(\boldsymbol{\theta})}{\partial \bar{\omega}.(\cdot)}\Big| \;\le\; \sum_{y\in\mathcal{V}}\sum_{\tilde{\mathbf{s}}\sqsubseteq\mathbf{s}}\big|\Delta_{\mathcal{L}(\boldsymbol{\theta})}(\tilde{\mathbf{s}},y)\big|.$$

$\square$

## B.3 Decoder-only Transformer as Subsequence Embeddor

**Overview.** The decoder-only Transformer architecture represents input sequences as hierarchical feature matrices through multiple layers. At the $l$-th layer, the feature matrix $\Phi^{(l)}(\mathbf{s}) \in \mathbb{R}^{t \times d}$ aggregates feature vectors for the input sequence $\mathbf{s} = [x_1, x_2, \ldots, x_n]$. At the first layer, the feature matrix is given by the stacking of token embedding:

$$\Phi^{(1)}(\mathbf{s}) = [\varphi(x_1), \varphi(x_2), \ldots, \varphi(x_n)]^\top \in \mathbb{R}^{n \times d}.$$

The computation of $\Phi^{(l)}(\mathbf{s})$ at layer $l$ is defined recursively as:

$$\Phi^{(l)}(\mathbf{s}) = \Phi^{(l-1)}(\mathbf{s}) + \text{FFN}^{(l-1)}\left(\text{MHA}^{(l-1)}\left(\Phi^{(l-1)}(\mathbf{s})\right)\right),$$

where $\text{MHA}^{(l)}$ is the multi-head attention block, and $\text{FFN}^{(l)}$ is the feed-forward layer at the $l$-th layer. As a side note, the feature $\Phi^{(l)}(\mathbf{s}) \in \mathbb{R}^{n \times d}$ can be decomposed into $n$ vectors:

$$\Phi^{(l)}(\mathbf{s}) = [\phi_1^{(l)}(\mathbf{s}), \phi_2^{(l)}(\mathbf{s}), ..., \phi_n^{(l)}(\mathbf{s})]^\top,$$

**Multi-head Attention Block.** The multi-head attention (MHA) block is a core component of the Transformer, enabling the model to focus on relevant parts of the input sequence. At the $l$-th layer, the MHA block computes an output feature matrix as the concatenation of outputs from $H$ attention heads:

$$\text{MHA}^{(l)}(\Phi) = \text{concat}\left(\text{MHA}^{(l,1)}(\Phi), \text{MHA}^{(l,2)}(\Phi), \ldots, \text{MHA}^{(l,H)}(\Phi)\right) \in \mathbb{R}^{Hd}.$$

For each attention head $h \in \{1, \ldots, H\}$, the output $\text{MHA}^{(l,h)}(\Phi)$ is computed as:

$$\text{MHA}^{(l,h)}(\Phi) = \mathbf{W}_{OV}^{(l,h)}\Phi^\top \cdot \boldsymbol{\sigma}\left(\Phi \mathbf{W}_{KQ}^{(l,h)}\Phi_{-1,:}^\top\right) \in \mathbb{R}^d,$$

where:

- $\boldsymbol{\sigma}(\cdot)$ denotes the row-wise softmax function.

- $\Phi \in \mathbb{R}^{t \times d}$ is the feature matrix up to position $t$, and $\Phi_{-1,:}$ is the feature vector of the last token. The causal mask ensures that attention is restricted to the last token's feature.

- $\mathbf{W}_{KQ}^{(l,h)} \in \mathbb{R}^{d \times d}$ and $\mathbf{W}_{OV}^{(l,h)} \in \mathbb{R}^{d \times d}$ are learnable weight matrices for key-query and output-value transformations, respectively, defined as:

$$\mathbf{W}_{KQ}^{(l,h)} = \mathbf{W}_K^{(l,h)\top}\mathbf{W}_Q^{(l,h)}, \quad \mathbf{W}_{OV}^{(l,h)} = \mathbf{W}_O^{(l,h)\top}\mathbf{W}_V^{(l,h)}.$$

**Feed-forward Layer.** The feed-forward network (FFN) layer applies a non-linear transformation to the input feature vectors. At the $l$-th layer, it is defined as:

$$\text{FFN}^{(l)}(\mathbf{v}) = W_{F_2}^{(l)}\text{ReLU}\left(W_{F_1}^{(l)}\mathbf{v} - \mathbf{b}^{(l)}\right),$$

where:

- $W_{F_1}^{(l)} \in \mathbb{R}^{d \times d_f}$ and $W_{F_2}^{(l)} \in \mathbb{R}^{d_f \times Hd}$ are learnable weight matrices.

- $\mathbf{b}^{(l)} \in \mathbb{R}^{d_f}$ is the bias vector.

- $\mathbf{v} \in \mathbb{R}^{Hd}$ is typically the output of the multi-head attention block.

This layer enhances the representational capacity of the model through non-linear activation and dimensionality transformations.

**Lemma B.2.** *For any length-2 subsequence $[\tilde{x}, \tilde{x}']$ within an input sequence $\mathbf{s}$ with length $n \geq 2$, a single-layer transformer block can act as an embedding function for subsequence: $\varphi([\tilde{x}, \tilde{x}']) \cdot \mathbf{1}\{[\tilde{x}, \tilde{x}'] \sqsubseteq \mathbf{s}\}$.*

*Proof.* Consider a single-layer multi-head attention block with two attention heads. The output feature at position $t$ is the concatenation of the outputs from each head:

$$\text{MHA}^{(1,:)}(\Phi_{[:t]}^{(1)}(\mathbf{s})) = \text{concat}\left(\text{MHA}^{(1,1)}(\Phi_{[:t]}^{(1)}(\mathbf{s})), \text{MHA}^{(1,2)}(\Phi_{[:t]}^{(1)}(\mathbf{s}))\right).$$

For each head $h \in \{1, 2\}$, the output $\text{MHA}^{(1,h)}(\Phi_{[:t]}^{(1)}(\mathbf{s}))$ is computed as:

$$\text{MHA}^{(1,h)}(\Phi_{[:t]}^{(1)}(\mathbf{s})) = \mathbf{W}_{OV}^{(1,h)}\Phi_{[:t]}^{(1)}(\mathbf{s})^\top \cdot \boldsymbol{\sigma}\left(\Phi_{[:t]}^{(1)}(\mathbf{s})\mathbf{W}_{KQ}^{(1,h)}\varphi(x_t)\right),$$

where:

- $\varphi(x_t) \in \mathbb{R}^d$ is the embedding of the token at position $t$.

- $\Phi_{[:t]}^{(1)}(\mathbf{s}) \in \mathbb{R}^{t \times d}$ is the embedding matrix for the sequence $\mathbf{s}$ up to position $t$, i.e., $\Phi_{[:t]}^{(1)}(\mathbf{s}) = [\varphi(x_1), \varphi(x_2), \dots, \varphi(x_t)]^\top$.

To configure the transformer to select a specific subsequence $[\tilde{x}, \tilde{x}']$, the weight matrices are constructed as:

$$\mathbf{W}_{KQ}^{(1,1)} = \gamma\varphi(\tilde{x})\varphi(\tilde{x}')^\top, \quad \mathbf{W}_{OV}^{(1,1)} = \mathbf{W}_{KQ}^{(1,2)} = \mathbf{W}_{OV}^{(1,2)} = \mathbf{I}_{d \times d},$$

where $\mathbf{I}_{d \times d}$ is the identity matrix and $\gamma \gg 1$.

With these weights, in the feed-forward network (FFN) layer following the attention block, a linear filter can serve as the indicator of the subsequence $[\tilde{x}, \tilde{x}']$:

$$\mathbf{1}\left\{[\tilde{x}, \tilde{x}'] \sqsubseteq \mathbf{s}\right\} \approx \max_{t=1,\dots,n} \text{ReLU}\left(\langle \mathbf{w}_{F_1}, \text{MHA}^{(1,:)}(\Phi_{[:t]}^{(1)}(\mathbf{s}))\rangle - b\right),$$

where:

- $\mathbf{w}_{F_1} = \text{concat}(\varphi(\tilde{x}), \varphi(\tilde{x}'))$,

- $b = 1$ is a bias term.

To see this, we first consider when $[\tilde{x}, \tilde{x}'] \sqsubseteq \mathbf{s}$ and we assume $\tilde{x}'$ appear at the position $j$. We have:

$$\text{MHA}^{(1,1)}(\Phi_{[:j]}^{(1)}(\mathbf{s})) = (1 - O\left(e^{\gamma(O(\epsilon)-1)}\right))\varphi(\tilde{x}) + (O\left(e^{\gamma(O(\epsilon)-1)}\right))\sum_{i=1}^{j}\varphi(x_i)\mathbf{1}\{x_i \neq \tilde{x}\}$$

$$= \varphi(\tilde{x}) + O\left(e^{\gamma(O(\epsilon)-1)}\right)$$

and similarly,

$$\text{MHA}^{(1,2)}(\Phi_{[:j]}^{(1)}(\mathbf{s})) = \varphi(\tilde{x}') + O\left(e^{\gamma(O(\epsilon)-1)}\right),$$

ensuring that the concatenated output at position $j$ is:

$$\text{MHA}^{(1,:)}(\Phi_{[:j]}^{(1)}(\mathbf{s})) = \text{concat}(\varphi(\tilde{x}), \varphi(\tilde{x}')) + O\left(e^{\gamma(O(\epsilon)-1)}\right).$$

The ReLU output will thus be approximately 1. On the contrary, if $[\tilde{x}, \tilde{x}'] \not\sqsubseteq \mathbf{s}$, then $\text{MHA}^{(1,1)}(\Phi_{[:j]}^{(1)}(\mathbf{s}))$ becomes a random linear combination of $\varphi(x)$ that $x \neq \tilde{x}$. In this case, the ReLU activation output is most $O\left(e^{\gamma(O(\epsilon)-1)}\right) \approx 0, (\gamma \gg 1)$, even if $\tilde{x}' \in \mathbf{s}$. Suppose our FFN only has one hidden neuron ($d_f = 1$), the output of FFN serves as the embedding of the subsequence $[\tilde{x}, \tilde{x}']$ if we let $\mathbf{w}_{F_2} = \varphi([\tilde{x}, \tilde{x}'])$:

$$\text{FFN}^{(l)}(\mathbf{v}) = \sum_{t=1}^{n}\mathbf{w}_{F_2}\text{ReLU}\left(\langle \mathbf{w}_{F_1}, \text{MHA}^{(1,:)}(\Phi_{[:t]}^{(1)}(\mathbf{s}))\rangle - b\right) \approx \varphi([\tilde{x}, \tilde{x}']) \cdot \mathbf{1}\left\{[\tilde{x}, \tilde{x}'] \sqsubseteq \mathbf{s}\right\}.$$

$\square$

The next Lemma serves as a generalization of the Lemma B.2 with more than one length-2 subsequence.

**Lemma B.3.** *For a set of length-2 subsequence* $\mathbf{S}^2 = \{\tilde{\mathbf{s}}_i\}_{i=1}^{N_s}$ *with* $\tilde{\mathbf{s}}_i = [\tilde{x}_i, \tilde{x}_i']$, *a single-layer transformer block with* $d_f = N_s, d = O\left(\epsilon^{-2}\ln\left(\epsilon(|\mathcal{V}| + N_s)\right)\right)$ *can act as an embedding function* $\varphi : \mathcal{V}^2 \mapsto \mathbb{R}^d$ *for all the length-2 subsequence in* $\mathbf{S}^2$, *ensuring that*

$$\forall \tilde{\mathbf{s}}, \tilde{\mathbf{s}}' \in \mathbf{S}^2, x \in \mathcal{V}, |\langle \varphi(\tilde{\mathbf{s}}), \varphi(x)\rangle| < \epsilon, |\langle \varphi(\tilde{\mathbf{s}}), \varphi(\tilde{\mathbf{s}}')\rangle| < \epsilon$$

*for a small* $\epsilon > 0$. *Specifically, for an input sequence* $\mathbf{s} = [x_1, x_2, ..., x_n] \in \mathcal{V}^n$, *the embedded results* $\Phi^{(2)}(\mathbf{s}) \in \mathbb{R}^{n \times d}$ *whose would be*

$$\Phi^{(2)}(\mathbf{s}) = [\phi_1^{(2)}(\mathbf{s}), \phi_2^{(2)}(\mathbf{s}), ..., \phi_n^{(2)}(\mathbf{s})]^\top,$$

*where*

$$\phi_t^{(2)}(\mathbf{s}) = \sum_{\{\tilde{\mathbf{s}}|\tilde{\mathbf{s}}\sqsubseteq\mathbf{S}^2, \tilde{\mathbf{s}}\sqsubseteq\mathbf{s}_{[:t]}, \tilde{\mathbf{s}}_{[2]}=x_t\}} \mu_{\tilde{\mathbf{s}}} \cdot \varphi(\tilde{\mathbf{s}}), \text{with } \mu_{\tilde{\mathbf{s}}} > 0.$$

*Proof.* We construct a single-layer, two-head multi-head attention (MHA) block, followed by a feed-forward network (FFN) of hidden dimension $d_f = N_s$, as follows. For MHA block, let

$$\mathrm{MHA}^{(1,:)}(\Phi_{[:t]}^{(1)}(\mathbf{s})) = \mathrm{concat}\left(\mathrm{MHA}^{(1,1)}(\Phi_{[:t]}^{(1)}(\mathbf{s})), \mathrm{MHA}^{(1,2)}(\Phi_{[:t]}^{(1)}(\mathbf{s}))\right),$$

where

$$\mathrm{MHA}^{(1,h)}(\Phi_{[:t]}^{(1)}(\mathbf{s})) = \mathbf{W}_{OV}^{(1,h)}\,\Phi_{[:t]}^{(1)}(\mathbf{s})^\top \cdot \boldsymbol{\sigma}\left(\Phi_{[:t]}^{(1)}(\mathbf{s})\,\mathbf{W}_{KQ}^{(1,h)}\,\varphi(x_t)\right), \quad h \in \{1, 2\}.$$

Here, $\Phi_{[:t]}^{(1)}(\mathbf{s}) \in \mathbb{R}^{t \times d}$ collects embeddings $\varphi(x_1), \ldots, \varphi(x_t)$ up to position $t$. Following Lemma B.2, we assign:

$$\mathbf{W}_{KQ}^{(1,1)} = \sum_{[\tilde{x},\tilde{x}']\in\mathbf{S}^2} \gamma_{\tilde{\mathbf{s}}}\,\varphi(\tilde{x})\,\varphi(\tilde{x}')^\top, \quad \mathbf{W}_{OV}^{(1,1)} = \mathbf{W}_{KQ}^{(1,2)} = \mathbf{W}_{OV}^{(1,2)} = \mathbf{I}_{d\times d},$$

where each $\gamma_{\tilde{\mathbf{s}}} \gg 1$ is a large scalar chosen per subsequence $\tilde{\mathbf{s}} \in \mathbf{S}^2$.

*Key Intuition.* As in the proof of Lemma B.2, multiplying $\varphi(x_t)$ by $\mathbf{W}_{KQ}^{(1,1)}$ extracts large positive activation only when $\varphi(x_t) = \varphi(\tilde{x}')$ *and* there exists an $\tilde{x}$ in the preceding positions that aligns with $\varphi(\tilde{x})$. This ensures that whenever $\tilde{x}$ appears somewhere in $\mathbf{s}$ before $t$ and $\tilde{x}'$ appears at position $t$, the attention head 1 "detects" this subsequence, yielding an output for all the $\tilde{x}$ satisfying the condition:

$$\mathrm{MHA}^{(1,1)}(\Phi_{[:t]}^{(1)}(\mathbf{s})) = \sum_{\tilde{x}\in\{x|[x,x_t]\sqsubseteq\mathbf{s}_{[:t]}, [x,x_t]\in\mathbf{S}^2\}} \mu_{[\tilde{x},x_t]} \cdot \varphi(\tilde{x}),$$

where $\mu_{[\tilde{x},x_t]}$ is a positive number dependent on the occurrence of the $\tilde{x}$ preceding and the $\gamma_{[\tilde{x},x_t]}$. If $[\tilde{x}, \tilde{x}'] \in \mathbf{S}^2$ and $[\tilde{x}, \tilde{x}'] \sqsubseteq \mathbf{s}_{[:t]}$,

$$\mu_{[\tilde{x},x_t]} = \frac{\exp(C(\tilde{x}, \mathbf{s}_{[:t]}) \cdot \gamma_{[\tilde{x},x_t]})}{\displaystyle\sum_{x\in\{x|[x,x_t]\sqsubseteq\mathbf{s}_{[:t]}, [x,x_t]\in\mathbf{S}^2\}} \exp(C(x, \mathbf{s}_{[:t]}) \cdot \gamma_{[x,x_t]}) + \sum_{x\in\{x|[x,x_t]\sqsubseteq\mathbf{s}_{[:t]}, [x,x_t]\notin\mathbf{S}^2\}} \exp(C(x, \mathbf{s}_{[:t]}) \cdot O(\epsilon)}.$$

where $C(x, \mathbf{s}_{[:t]})$ denotes the number of $x$ that occurs in the first-$t$-token subsequence $\mathbf{s}_{[:t]}$. In the rest of the proof, we will omit the calculation details for the softmax expansion.

Meanwhile, attention head 2 simply copies the token embedding at the current position (because $\mathbf{W}_{KQ}^{(1,2)} = \mathbf{I}$), thus producing $\varphi(\tilde{x}')$. Concatenating these two heads gives an output at position $t$ that is approximately

$$\mathrm{concat}\left(\sum_{\tilde{x}\in\{x|[x,x_t]\sqsubseteq\mathbf{s}_{[:t]}, [x,x_t]\in\mathbf{S}^2\}} \mu_{[\tilde{x},x_t]} \cdot \varphi(\tilde{x}), \varphi(\tilde{x}')\right),$$

if and only if those subsequence $[\tilde{x}, \tilde{x}']$ occurs ending at position $t$. Otherwise, the first part of vector is near zero or some negligible combination (due to large $\gamma$), as in Lemma B.2.

Next, we feed the MHA output into a single-layer FFN with $N_s$ hidden neurons to encode each $\widetilde{\mathbf{s}}_i \in \mathbf{S}^2$. Concretely, let

$$\text{FFN}^{(1)}(\mathbf{v}) \;=\; \mathbf{W}_{F_2}\,\text{ReLU}\Big(\big\langle \mathbf{W}_{F_1}, \mathbf{v}\big\rangle \;-\; \mathbf{b}\Big),$$

where $\mathbf{v} \in \mathbb{R}^{2d}$ is the concatenated output $\text{MHA}^{(1,:)}(\cdot)$. We set:

$$\mathbf{W}_{F_1} = \begin{bmatrix} \text{concat}(\varphi(\tilde{x}_1), \varphi(\tilde{x}'_1))^\top \\ \text{concat}(\varphi(\tilde{x}_2), \varphi(\tilde{x}'_2))^\top \\ \vdots \\ \text{concat}(\varphi(\tilde{x}_{N_s}), \varphi(\tilde{x}'_{N_s}))^\top \end{bmatrix} \in \mathbb{R}^{N_s \times 2d}, \quad \mathbf{W}_{F_2} = \big[\varphi(\widetilde{\mathbf{s}}_1),\, \varphi(\widetilde{\mathbf{s}}_2),\, \dots,\, \varphi(\widetilde{\mathbf{s}}_{N_s})\big] \in \mathbb{R}^{d \times N_s},$$

and $\mathbf{b} = \mathbf{1}_{N_s} \in \mathbb{R}^{N_s}$.

Observe that for the $i$-th row in $\mathbf{W}_{F_1}$, the ReLU activation

$$\text{ReLU}\big(\langle \text{concat}(\text{MHA}^{(1,:)}(\Phi^{(1)}_{[:t]}(\mathbf{s})), \varphi(\tilde{x}'_i)),\, \text{concat}(\varphi(\tilde{x}), \varphi(\tilde{x}'))\rangle - 1\big) \approx$$

$$\sum_{\tilde{x} \in \{x | [x, x_t] \sqsubseteq \mathbf{s}_{[:t]}, [x, x_t] \in \mathbf{S}^2\}} \text{ReLU}\big(\langle \text{concat}(\mu_{[\tilde{x}_i, \tilde{x}'_i]} \cdot \varphi(\tilde{x}_i), \varphi(\tilde{x}'_i)),\, \text{concat}(\varphi(\tilde{x}), \varphi(\tilde{x}'))\rangle - 1\big) = \mu_{[\tilde{x}_i, \tilde{x}'_i]} + O(\epsilon)$$

if $\text{concat}(\varphi(\tilde{x}), \varphi(\tilde{x}')) = \text{concat}(\varphi(\tilde{x}_i), \varphi(\tilde{x}'_i))$, and near 0 otherwise. Hence neuron $i$ "fires" if and only if subsequence $[\tilde{x}_i, \tilde{x}'_i]$ is detected at the current position.

Multiplying by $\mathbf{W}_{F_2}$ (of size $d \times N_s$) gives

$$\phi^{(2)}_t(\mathbf{s}) = \text{FFN}^{(1)}\left(\text{MHA}^{(1)}\left(\Phi^{(1)}_{[:t]}(\mathbf{s})\right)\right) = \sum_{\{\tilde{\mathbf{s}} | \tilde{\mathbf{s}} \sqsubseteq \mathbf{S}^2, \tilde{\mathbf{s}} \sqsubseteq \mathbf{s}_{[:t]}, \tilde{\mathbf{s}}_{[2]} = x_t\}} \mu_{\tilde{\mathbf{s}}} \cdot \varphi(\tilde{\mathbf{s}}), \text{ with } \mu_{\tilde{\mathbf{s}}} > 0,$$

up to small error terms of order $O\big(e^{\gamma_{\tilde{\mathbf{s}}}(O(\epsilon)-1)}\big)$ as $\gamma_{\widetilde{\mathbf{s}}} \gg 1$ for all $\tilde{\mathbf{s}}$.

Finally, we ensure (1) $\langle \varphi(\widetilde{\mathbf{s}}), \varphi(x)\rangle \approx 0$ for all $x \in \mathcal{V}$ and (2) $\langle \varphi(\widetilde{\mathbf{s}}_i), \varphi(\widetilde{\mathbf{s}}_j)\rangle \approx 0$ for $\widetilde{\mathbf{s}}_i \neq \widetilde{\mathbf{s}}_j$.

Using the results in Theorem B.4, we are able to construct $\{\varphi(\widetilde{\mathbf{s}})\}_{\widetilde{\mathbf{s}} \in \mathbf{S}^2}$ and $\{\varphi(x)\}_{x \in \mathcal{V}}$ in $\mathbb{R}^d$ with $d = O\big(\epsilon^{-2}\ln(\epsilon(|\mathcal{V}| + N_s))\big)$ to satisfy the required near-orthogonal bounds. $\qquad\qquad\square$

The next theory serves as a generalization of the Lemma B.3 with more than one length-2 subsequence.

**Theorem B.4.** *For a set of subsequence $\mathcal{S}^{[:\eta]} = \{\tilde{\mathbf{s}}_i\}_{i=1}^{N_s} \cup \mathcal{V}$ with the length of $\tilde{\mathbf{s}}_i$ ranges from 2 to $\eta$, an $L$-layer transformer network with $L = O(\log_2(\eta)), d_f = O(N_s\eta), d = O\left(\epsilon^{-2}\ln\left(\epsilon(|\mathcal{V}| + N_s\eta)\right)\right)$ can act as an embedding function $\varphi : \cup_{i=1}^{\eta}\mathcal{V}^i \mapsto \mathbb{R}^d$ for all the subsequences in $\mathcal{S}^{[:\eta]}$, ensuring that*

$$\forall \tilde{\mathbf{s}}, \tilde{\mathbf{s}}' \in \mathcal{S}^{[:\eta]}, |\langle \varphi(\tilde{\mathbf{s}}), \varphi(\tilde{\mathbf{s}}')\rangle| < \epsilon$$

*for a small $\epsilon > 0$. Specifically, for an input sequence $\mathbf{s} = [x_1, x_2, ..., x_n] \in \mathcal{V}^n$, the embedded results $\Phi^{(l)}(\mathbf{s}) \in \mathbb{R}^{n \times d}$ whose would be*

$$\Phi^{(l)}(\mathbf{s}) = [\phi^{(l)}_1(\mathbf{s}), \phi^{(l)}_2(\mathbf{s}), ..., \phi^{(l)}_n(\mathbf{s})]^\top,$$

*where*

$$\phi^{(l)}_t(\mathbf{s}) = \sum_{\mathcal{S}^{[:\eta]} \cap Sub(\mathbf{s}, t)} \mu^l_{\tilde{\mathbf{s}}} \cdot \varphi(\tilde{\mathbf{s}}), \text{ with } \mu^l_{\tilde{\mathbf{s}}} > 0.$$

*Proof.* We prove the theorem by constructing, layer by layer, a Transformer whose outputs encode the near-orthogonal embeddings of all subsequences in $\mathcal{S}^{[:\eta]} = \{\tilde{\mathbf{s}}_i\}_{i=1}^{N_s} \cup \mathcal{V}$. The key idea is that a subsequence of length $\eta$ can be split into two parts—one of length roughly $\eta/2$ and the other of length $\eta/2$. Repeating this splitting procedure across multiple layers (on the order of $\log_2(\eta)$) allows the Transformer to recognize and embed all such subsequences.

*a) Decomposition of Subsequences.* We start by defining the decomposition of subsequences used in each layer:

1. Let $\mathcal{S}_L^{[:\eta]} = \mathcal{S}^{[:\eta]}$ denote the set of all subsequences (of length up to $\eta$) at the final layer $L$.

2. At each layer $l \in \{1, 2, \ldots, L\}$, every subsequence $\widetilde{\mathbf{s}} \in \mathcal{S}_L^{[:\eta]}$ is split into two parts:

$$\widetilde{\mathbf{s}} = [\widetilde{\mathbf{s}}_\triangleleft, \widetilde{\mathbf{s}}_\triangleright],$$

where $\mathrm{splitleft}(\widetilde{\mathbf{s}}) = \widetilde{\mathbf{s}}_\triangleleft$ and $\mathrm{splitright}(\widetilde{\mathbf{s}}) = \widetilde{\mathbf{s}}_\triangleright$. The left and right parts satisfy $|\widetilde{\mathbf{s}}_\triangleleft| \geq |\widetilde{\mathbf{s}}_\triangleright|$.

3. The set $\mathcal{S}_{l-1}^{[:\eta]}$ is then the collection of all such left and right parts of subsequences from $\mathcal{S}_L^{[:\eta]}$:

$$\mathcal{S}_{l-1}^{[:\eta]} \;=\; \big\{\, \widetilde{\mathbf{s}}_\triangleleft \mid \widetilde{\mathbf{s}} \in \mathcal{S}_L^{[:\eta]} \big\} \,\cup\, \big\{\, \widetilde{\mathbf{s}}_\triangleright \mid \widetilde{\mathbf{s}} \in \mathcal{S}_L^{[:\eta]} \big\}.$$

For example, if $\widetilde{\mathbf{s}} = [a, b, c]$, then:

$$\widetilde{\mathbf{s}}_\triangleleft = [a, b], \quad \widetilde{\mathbf{s}}_\triangleright = [c].$$

If $\widetilde{\mathbf{s}} = [a]$ (length 1), we pad the right part as $\varnothing$. These splits allow the Transformer to "build up" embeddings of longer subsequences by combining embeddings of their shorter left and right parts across successive layers.

*b) Inductive Construction of the Transformer Output.* Let $\Phi^{(l)}(\mathbf{s}, n) \in \mathbb{R}^{n \times d}$ be the output of the $l$-th layer of the Transformer, where the $t$-th row is denoted as

$$\phi_t^{(l)}(\mathbf{s}) \;\in\; \mathbb{R}^d, \quad t = 1, \ldots, n.$$

We assume an inductive format for $\phi_t^{(l)}(\mathbf{s})$, namely:

$$\phi_t^{(l)}(\mathbf{s}) \;=\; \sum_{\{\widetilde{\mathbf{s}}' \mid \widetilde{\mathbf{s}}' \in \mathcal{S}_L^{[:\eta]},\, \widetilde{\mathbf{s}}' \sqsubseteq \mathbf{s}_{[:t]},\, \widetilde{\mathbf{s}}'_{[-1]} = x_t\}} \mu_{\widetilde{\mathbf{s}}'}\, \varphi(\widetilde{\mathbf{s}}'),$$

where $\widetilde{\mathbf{s}}' \sqsubseteq \mathbf{s}_{[:t]}$ denotes that $\widetilde{\mathbf{s}}'$ is a subsequence of $\mathbf{s}$ ending at position $t$, with last element $x_t$. The scalar weights $\mu_{\widetilde{\mathbf{s}}'} > 0$ ensure positivity in the linear combination.

We prove by induction on $l$ that:

$$\phi_t^{(l+1)}(\mathbf{s}) \;=\; \sum_{\{\widetilde{\mathbf{s}} \mid \widetilde{\mathbf{s}} \in \mathcal{S}_{l+1}^{[:\eta]},\, \widetilde{\mathbf{s}} \sqsubseteq \mathbf{s}_{[:t]},\, \widetilde{\mathbf{s}}_{[-1]} = x_t\}} \mu_{\widetilde{\mathbf{s}}}\, \varphi(\widetilde{\mathbf{s}}).$$

*b-1) Base Case $l = 1$.*

By Lemma B.3, after applying the first layer's multi-head attention (MHA) and feed-forward network (FFN), we have:

$$\mathrm{FFN}^{(1)}\Big(\mathrm{MHA}^{(1)}\big(\Phi^{(1)}(\mathbf{s}, n)\big)\Big)^\top\Big|_{t\text{-th column}} = \sum_{\{\widetilde{\mathbf{s}} \in \mathbf{s}^2 \mid \widetilde{\mathbf{s}} \sqsubseteq \mathbf{s}_{[:t]},\, \widetilde{\mathbf{s}}_{[2]} = x_t\}} \mu_{\widetilde{\mathbf{s}}}\, \varphi(\widetilde{\mathbf{s}}),$$

where $\varphi(\widetilde{\mathbf{s}})$ is an embedding for 2-element subsequences, and $\mu_{\widetilde{\mathbf{s}}} > 0$. Incorporating a routing/residual layer yields:

$$\phi_t^{(2)}(\mathbf{s}) = \phi_t^{(1)}(\mathbf{s}) + \sum_{\{\widetilde{\mathbf{s}} \in \mathbf{s}^2 \mid \widetilde{\mathbf{s}} \sqsubseteq \mathbf{s}_{[:t]},\, \widetilde{\mathbf{s}}_{[2]} = x_t\}} \mu_{\widetilde{\mathbf{s}}}\, \varphi(\widetilde{\mathbf{s}}).$$

Since $\varphi(\cdot)$ and $\varphi(\cdot)$ can be unified into a single function $\varphi(\cdot)$ mapping to the same $d$-dimensional space, we obtain

$$\phi_t^{(2)}(\mathbf{s}) = \sum_{\{\widetilde{\mathbf{s}} \in (\mathbf{S}^2 \cup \mathcal{V}) \mid \widetilde{\mathbf{s}} \sqsubseteq \mathbf{s}_{[:t]},\, \widetilde{\mathbf{s}}_{[-1]} = x_t\}} \mu_{\widetilde{\mathbf{s}}}\, \varphi(\widetilde{\mathbf{s}}),$$

matching the claimed inductive form for $l = 1$.

*b-2) Inductive Step.* Assume the form holds at layer $l$. We show it remains valid at layer $l + 1$.

*Multi-Head Attention Construction.* We use two attention heads at layer $l$:

$$\mathrm{MHA}^{(l,:)}\big(\Phi_{[:t]}^{(l)}(\mathbf{s})\big) = \mathrm{concat}\Big(\mathrm{MHA}^{(l,1)}(\Phi_{[:t]}^{(l)}(\mathbf{s})),\, \mathrm{MHA}^{(l,2)}(\Phi_{[:t]}^{(l)}(\mathbf{s}))\Big),$$

where $\Phi_{[:t]}^{(l)}(\mathbf{s}) \in \mathbb{R}^{t \times d}$ collects all row-embeddings $\phi_1^{(l)}(\mathbf{s}), \dots, \phi_t^{(l)}(\mathbf{s})$. Each head is defined by:

$$\text{MHA}^{(l,h)}\big(\Phi_{[:t]}^{(l)}(\mathbf{s})\big) = \mathbf{W}_{OV}^{(l,h)} \, \Phi_{[:t]}^{(l)}(\mathbf{s})^\top \cdot \boldsymbol{\sigma}\Big(\Phi_{[:t]}^{(l)}(\mathbf{s}) \, \mathbf{W}_{KQ}^{(l,h)} \, \phi_t^{(l)}(\mathbf{s})\Big), \quad h \in \{1, 2\}.$$

Similar to LemmasB.2 and B.3, choose

$$\mathbf{W}_{KQ}^{(l,1)} = \sum_{\widetilde{\mathbf{s}} = [\widetilde{\mathbf{s}}_\lhd, \widetilde{\mathbf{s}}_\rhd] \in \mathcal{S}_{l+1}^{[:\eta]}} \gamma_{\widetilde{\mathbf{s}}} \, \varphi(\widetilde{\mathbf{s}}_\lhd) \, \varphi(\widetilde{\mathbf{s}}_\rhd)^\top, \quad \mathbf{W}_{OV}^{(l,1)} = \mathbf{W}_{KQ}^{(l,2)} = \mathbf{W}_{OV}^{(l,2)} = \mathbf{I}_{d \times d},$$

where $\gamma_{\widetilde{\mathbf{s}}} \gg 1$ is chosen large enough that attention "activates" primarily when $\phi_i^{(l)}(\mathbf{s})$ contains $\varphi(\widetilde{\mathbf{s}}_\lhd)$ and $\phi_t^{(l)}(\mathbf{s})$ contains $\varphi(\widetilde{\mathbf{s}}_\rhd)$.

*Head 1* detects occurrences of the pair $\big(\widetilde{\mathbf{s}}_\lhd, \widetilde{\mathbf{s}}_\rhd\big)$ across indices $\{1, \dots, t\}$. Concretely,

$$\text{MHA}^{(l,1)}\big(\Phi_{[:t]}^{(l)}(\mathbf{s})\big) = \sum_{\{\widetilde{\mathbf{s}} \,|\, \widetilde{\mathbf{s}} = [\widetilde{\mathbf{s}}_\lhd, \widetilde{\mathbf{s}}_\rhd] \in \mathcal{S}_{l+1}^{[:\eta]}, \, \widetilde{\mathbf{s}} \sqsubseteq \mathbf{s}_{[:t]}, \, \widetilde{\mathbf{s}}_{\rhd[-1]} = x_t\}} \mu_{\widetilde{\mathbf{s}}} \, \varphi(\widetilde{\mathbf{s}}_\lhd).$$

*Head 2* simply copies the $\phi_t^{(l)}(\mathbf{s})$ feature from the previous layer:

$$\text{MHA}^{(l,2)}\big(\Phi_{[:t]}^{(l)}(\mathbf{s})\big) = \phi_t^{(l)}(\mathbf{s}) = \sum_{\{\widetilde{\mathbf{s}}_\rhd \in \mathcal{S}_L^{[:\eta]} \,|\, \widetilde{\mathbf{s}}_\rhd \sqsubseteq \mathbf{s}_{[:t]}, \, \widetilde{\mathbf{s}}_{\rhd[-1]} = x_t\}} \mu_{\widetilde{\mathbf{s}}_\rhd} \, \varphi(\widetilde{\mathbf{s}}_\rhd).$$

*Feed-Forward Network (FFN).* We next pass the concatenated output $\mathbf{v} \in \mathbb{R}^{2d}$ from these two heads into an FFN:

$$\text{FFN}^{(l)}(\mathbf{v}) = \mathbf{W}_{F_2}^{(l)} \, \sigma\Big(\mathbf{W}_{F_1}^{(l)} \mathbf{v} - \mathbf{b}^{(l)}\Big),$$

where $\sigma(\cdot)$ is (for instance) ReLU, and

$$\mathbf{W}_{F_1}^{(l)} = \Big[\text{concat}\big(\varphi(\widetilde{\mathbf{s}}_\lhd), \, \varphi(\widetilde{\mathbf{s}}_\rhd)\big)^\top \,:\, \widetilde{\mathbf{s}} = [\widetilde{\mathbf{s}}_\lhd, \widetilde{\mathbf{s}}_\rhd] \in \mathcal{S}_{l+1}^{[:\eta]}\Big], \quad \mathbf{W}_{F_2}^{(l)} = \Big[\varphi(\widetilde{\mathbf{s}}) \,:\, \widetilde{\mathbf{s}} \in \mathcal{S}_{l+1}^{[:\eta]}\Big].$$

A suitable choice of bias $\mathbf{b}^{(l)}$ ensures that the corresponding neuron "fires" when the input $\text{concat}\big(\varphi(\widetilde{\mathbf{s}}_\lhd), \, \varphi(\widetilde{\mathbf{s}}_\rhd)\big)$ matches one of the rows in $\mathbf{W}_{F_1}^{(l)}$. Consequently, the output of the $l$-th layer FFN at position $t$ is:

$$\phi_t^{(l+1)}(\mathbf{s}) = \sum_{\{\widetilde{\mathbf{s}} \,|\, \widetilde{\mathbf{s}} \in \mathcal{S}_{l+1}^{[:\eta]}, \, \widetilde{\mathbf{s}} \sqsubseteq \mathbf{s}_{[:t]}, \, \widetilde{\mathbf{s}}_{[-1]} = x_t\}} \mu_{\widetilde{\mathbf{s}}}^{(l+1)} \, \varphi(\widetilde{\mathbf{s}}),$$

which completes the induction step.

*c) Near-Orthogonality of Embeddings.* Across all layers $l$, the total number of subsequences in $\bigcup_{l=2}^{O(\log_2(\eta))} \mathcal{S}_L^{[:\eta]}$ is on the order of $O(N_s \eta)$. By setting the embedding dimension

$$d = O\Big(\epsilon^{-2} \ln\big[\epsilon \, (|\mathcal{V}| + N_s \, \eta)\big]\Big),$$

we can construct a mapping $\varphi(\cdot) \colon \mathcal{S}^{[:\eta]} \to \mathbb{R}^d$ so that

$$\forall \, \widetilde{\mathbf{s}} \neq \widetilde{\mathbf{s}}' \quad \text{in} \quad \mathcal{S}^{[:\eta]}, \quad \big|\langle \varphi(\widetilde{\mathbf{s}}), \, \varphi(\widetilde{\mathbf{s}}') \rangle\big| < \epsilon.$$

These embeddings are thus approximately orthogonal, ensuring $\big|\langle \varphi(\widetilde{\mathbf{s}}), \, \varphi(\widetilde{\mathbf{s}}') \rangle\big| < \epsilon$.

Combining the above arguments completes the proof: an $L$-layer Transformer with $L = O(\log_2(\eta))$, hidden dimension $d_f = O(N_s \eta)$, and embedding dimension $d = O\big(\epsilon^{-2} \ln(\epsilon(|\mathcal{V}| + N_s \eta))\big)$ can embed all subsequences in $\mathcal{S}^{[:\eta]}$ while maintaining near-orthogonality among their embeddings.

$\square$

### B.4 Other Useful Results

**Log-probability factorization with subsequence association**

**Proposition B.5.** *Let $\mathbf{s}'$ be an input sequence, and let $\tilde{\mathbf{s}} \sqsubseteq \mathbf{s}'$ denote its subsequences. Assume that:*

1. *The subsequences $\tilde{\mathbf{s}} \sqsubseteq \mathbf{s}'$ are marginally independent;*

2. *The subsequences $\tilde{\mathbf{s}} \sqsubseteq \mathbf{s}'$ are conditionally independent given $\tilde{\mathbf{o}} \sqsubseteq \mathbf{o}$.*

*Then, the log-probability of hallucination tokens $\tilde{\mathbf{o}} \sqsubseteq \mathbf{o}$ conditioned on $\mathbf{s}'$ is given by:*

$$\mathbb{P}_{\mathbf{s}\sim\mathcal{P}_{\mathbf{s}},\mathbf{o}\sim F(\mathbf{s})}(\tilde{\mathbf{o}} \sqsubseteq \mathbf{o} \mid \mathbf{s} = \mathbf{s}') = \sum_{\tilde{\mathbf{s}}\sqsubseteq\mathbf{s}'} \Psi(\tilde{\mathbf{s}},\tilde{\mathbf{o}}) + \mathbb{P}_{\mathbf{s}\sim\mathcal{P}_{\mathbf{s}},\mathbf{o}\sim F(\mathbf{s})}(\tilde{\mathbf{o}} \sqsubseteq \mathbf{o}),$$

*where the subsequence association $\Psi(\tilde{\mathbf{s}},\tilde{\mathbf{o}})$ is defined as:*

$$\Psi(\tilde{\mathbf{s}},\tilde{\mathbf{o}}) = \log \frac{\mathbb{P}_{\mathbf{s}\sim\mathcal{P}_{\mathbf{s}},\mathbf{o}\sim F(\mathbf{s})}(\tilde{\mathbf{o}} \sqsubseteq \mathbf{o} \mid \tilde{\mathbf{s}} \sqsubseteq \mathbf{s})}{\mathbb{P}_{\mathbf{s}\sim\mathcal{P}_{\mathbf{s}},\mathbf{o}\sim F(\mathbf{s})}(\tilde{\mathbf{o}} \sqsubseteq \mathbf{o})}.$$

*Proof.* By the conditional independence assumption and the Baysian rule, the conditional probability $\mathbb{P}_{\mathbf{s}\sim\mathcal{P}_{\mathbf{s}},\mathbf{o}\sim F(\mathbf{s})}(\tilde{\mathbf{o}} \sqsubseteq \mathbf{o} \mid \mathbf{s} = \mathbf{s}')$ can be factorized as:

$$\mathbb{P}_{\mathbf{s}\sim\mathcal{P}_{\mathbf{s}},\mathbf{o}\sim F(\mathbf{s})}(\tilde{\mathbf{o}} \sqsubseteq \mathbf{o} \mid \mathbf{s} = \mathbf{s}') = \mathbb{P}_{\mathbf{s}\sim\mathcal{P}_{\mathbf{s}},\mathbf{o}\sim F(\mathbf{s})}(\tilde{\mathbf{o}} \sqsubseteq \mathbf{o}) \prod_{\tilde{\mathbf{s}}\sqsubseteq\mathbf{s}'} \frac{\mathbb{P}_{\mathbf{s}\sim\mathcal{P}_{\mathbf{s}},\mathbf{o}\sim F(\mathbf{s})}(\tilde{\mathbf{o}} \sqsubseteq \mathbf{o} \mid \tilde{\mathbf{s}} \sqsubseteq \mathbf{s})}{\mathbb{P}_{\mathbf{s}\sim\mathcal{P}_{\mathbf{s}},\mathbf{o}\sim F(\mathbf{s})}(\tilde{\mathbf{o}} \sqsubseteq \mathbf{o})}.$$

Taking the Logarithm Taking the logarithm of both sides, we have:

$$\mathbb{P}_{\mathbf{s}\sim\mathcal{P}_{\mathbf{s}},\mathbf{o}\sim F(\mathbf{s})}(\tilde{\mathbf{o}} \sqsubseteq \mathbf{o} \mid \mathbf{s} = \mathbf{s}') = \mathbb{P}_{\mathbf{s}\sim\mathcal{P}_{\mathbf{s}},\mathbf{o}\sim F(\mathbf{s})}(\tilde{\mathbf{o}} \sqsubseteq \mathbf{o}) + \sum_{\tilde{\mathbf{s}}\sqsubseteq\mathbf{s}'} \log \frac{\mathbb{P}_{\mathbf{s}\sim\mathcal{P}_{\mathbf{s}},\mathbf{o}\sim F(\mathbf{s})}(\tilde{\mathbf{o}} \sqsubseteq \mathbf{o} \mid \tilde{\mathbf{s}} \sqsubseteq \mathbf{s})}{\mathbb{P}_{\mathbf{s}\sim\mathcal{P}_{\mathbf{s}},\mathbf{o}\sim F(\mathbf{s})}(\tilde{\mathbf{o}} \sqsubseteq \mathbf{o})}.$$

Definition of Subsequence Association From Definition 2.2, the subsequence association is given by:

$$\Psi(\tilde{\mathbf{s}},\tilde{\mathbf{o}}) = \log \frac{\mathbb{P}_{\mathbf{s}\sim\mathcal{P}_{\mathbf{s}},\mathbf{o}\sim F(\mathbf{s})}(\tilde{\mathbf{o}} \sqsubseteq \mathbf{o} \mid \tilde{\mathbf{s}} \sqsubseteq \mathbf{s})}{\mathbb{P}_{\mathbf{s}\sim\mathcal{P}_{\mathbf{s}},\mathbf{o}\sim F(\mathbf{s})}(\tilde{\mathbf{o}} \sqsubseteq \mathbf{o})}.$$

Substituting this into the equation, we obtain:

$$\mathbb{P}_{\mathbf{s}\sim\mathcal{P}_{\mathbf{s}},\mathbf{o}\sim F(\mathbf{s})}(\tilde{\mathbf{o}} \sqsubseteq \mathbf{o} \mid \mathbf{s} = \mathbf{s}') = \mathbb{P}_{\mathbf{s}\sim\mathcal{P}_{\mathbf{s}},\mathbf{o}\sim F(\mathbf{s})}(\tilde{\mathbf{o}} \sqsubseteq \mathbf{o}) + \sum_{\tilde{\mathbf{s}}\sqsubseteq\mathbf{s}'} \Psi(\tilde{\mathbf{s}},\tilde{\mathbf{o}}).$$

Thus, the proposition is proven. $\qquad\square$

**Maximum number of normalized embeddings with angular constraints.**

**Theorem B.6.** *Let $\mathcal{S}^{d-1}$ denote the unit sphere in $\mathbb{R}^d$. The maximum number $N(\epsilon, d)$ of unit vectors $\{\mathbf{v}_1, \mathbf{v}_2, \ldots, \mathbf{v}_{N(\epsilon,d)}\} \subseteq \mathcal{S}^{d-1}$ such that the angle between any pair of vectors exceeds $\pi/2 - \epsilon$ is bounded above by*

$$N(\epsilon, d) \leq \frac{\text{Surface Area of } \mathcal{S}^{d-1}}{\text{Volume of a Spherical Cap of Radius } \pi/2 - \epsilon}.$$

*For large $d$, $N(\epsilon, d)$ grows approximately as*

$$N(\epsilon, d) = O\left(\epsilon d \exp\left(\frac{d\epsilon^2}{2}\right)\right).$$

*Similarly, to include $N$ vectors such that $\forall \mathbf{v}_i, \mathbf{v}_j, |\mathbf{v}_i^\top \mathbf{v}_j| < \epsilon$, the dimension size needs to grow as*

$$d(N, \epsilon) = O\left(\frac{\ln(N\epsilon)}{\epsilon^2}\right).$$

*Proof.* To determine the maximum number of unit vectors $\{\mathbf{v}_1, \mathbf{v}_2, \ldots, \mathbf{v}_{N(\epsilon,d)}\}$ such that $\theta_{ij} > \pi/2 - \epsilon$ for all $i \neq j$, we consider the geometric constraints:

1. Each vector corresponds to a point on the unit sphere $\mathcal{S}^{d-1}$ in $\mathbb{R}^d$. 2. The angular constraint $\theta_{ij} > \pi/2 - \epsilon$ implies that no two points can lie within a spherical cap of angular radius $\pi/2 - \epsilon$.

The volume of a single spherical cap with angular radius $\pi/2 - \epsilon$ is given by $V_{\text{cap}}(\pi/2 - \epsilon)$. The total surface area of $\mathcal{S}^{d-1}$ is given by

$$\text{Surface Area}(\mathcal{S}^{d-1}) = \frac{2\pi^{d/2}}{\Gamma\left(\frac{d}{2}\right)}.$$

To pack $N(\epsilon, d)$ spherical caps of radius $\pi/2 - \epsilon$ on $\mathcal{S}^{d-1}$ without overlap, the total volume of these caps cannot exceed the total surface area of the sphere. Thus, we have the inequality

$$N(\epsilon, d) \cdot V_{\text{cap}}(\pi/2 - \epsilon) \leq \text{Surface Area}(\mathcal{S}^{d-1}).$$

Rearranging, we find

$$N(\epsilon, d) \leq \frac{\text{Surface Area}(\mathcal{S}^{d-1})}{V_{\text{cap}}(\pi/2 - \epsilon)} = \frac{1}{\int_0^{\pi/2-\epsilon} \sin^{d-2}(\theta)\, d\theta}.$$

For small $\epsilon$, $\cos(\epsilon)$ approaches 1, and the cap volume becomes concentrated around the equatorial region of the sphere. This leads to the asymptotic behavior

$$N(\epsilon, d) = O\left(\epsilon d \exp\left(\frac{d\epsilon^2}{2}\right)\right),$$

with detailed proof in Lemma B.7. By Lemma B.8, we have

$$d(N, \epsilon) = O\left(\frac{1}{\epsilon^2} \ln(N\epsilon)\right).$$

$\square$

**Lemma B.7.** *Let $d \in \mathbb{N}$ be large, and $\epsilon > 0$ be small such that $A = \epsilon\sqrt{d} \gg 1$. Consider the integral*

$$I(d, \epsilon) = \int_0^{\frac{\pi}{2}-\epsilon} \sin^{d-2}(\theta)\, d\theta.$$

*Then, in the regime $\epsilon\sqrt{d} \gg 1$, the reciprocal of the integral scales as:*

$$N(d, \epsilon) = O\left(\epsilon d \exp\left(\frac{d\epsilon^2}{2}\right)\right).$$

*Proof.* Set $x = \frac{\pi}{2} - \theta$, so that $\sin\theta = \cos x$. The integral becomes:

$$I(d, \epsilon) = \int_\epsilon^{\frac{\pi}{2}} \cos^{d-2}(x)\, dx.$$

For large $d$, the dominant contribution to the integral comes from small values of $x$, where $\cos x \approx 1 - \frac{x^2}{2}$. Substituting this approximation, we have:

$$\cos^{d-2}(x) \approx \exp\left(-(d-2)\frac{x^2}{2}\right).$$

Thus, the integral becomes:

$$I(d, \epsilon) \approx \int_\epsilon^\infty \exp\left(-\frac{(d-2)x^2}{2}\right) dx.$$

Define $t = \sqrt{d-2}\,x$, so that $x = \frac{t}{\sqrt{d-2}}$ and $dx = \frac{dt}{\sqrt{d-2}}$. The integral becomes:

$$I(d, \epsilon) = \frac{1}{\sqrt{d-2}} \int_{\epsilon\sqrt{d-2}}^\infty \exp\left(-\frac{t^2}{2}\right) dt.$$

Set $A = \epsilon\sqrt{d-2} \approx \epsilon\sqrt{d}$. Then:

$$I(d,\epsilon) \approx \frac{1}{\sqrt{d-2}} \int_A^\infty \exp\left(-\frac{t^2}{2}\right) dt.$$

When $A \gg 1$, the integral $\int_A^\infty \exp(-t^2/2)\, dt$ is dominated by the Gaussian tail, which satisfies:

$$\int_A^\infty \exp\left(-\frac{t^2}{2}\right) dt \sim \frac{\exp\left(-\frac{A^2}{2}\right)}{A}.$$

Substituting this into the expression for $I(d,\epsilon)$, we have:

$$I(d,\epsilon) \sim \frac{1}{\sqrt{d-2}} \cdot \frac{\exp\left(-\frac{A^2}{2}\right)}{A}.$$

Using $A = \epsilon\sqrt{d-2} \approx \epsilon\sqrt{d}$, we get:

$$I(d,\epsilon) \sim \frac{1}{\sqrt{d}} \cdot \frac{1}{\epsilon\sqrt{d}} \exp\left(-\frac{d\epsilon^2}{2}\right) = \frac{1}{\epsilon d} \exp\left(-\frac{d\epsilon^2}{2}\right).$$

The reciprocal of $I(d,\epsilon)$ is:

$$\frac{1}{I(d,\epsilon)} \sim \epsilon d \exp\left(\frac{d\epsilon^2}{2}\right).$$

$\square$

**Lemma B.8** (Growth of $d$ for Fixed $\epsilon$ and $N$). *Let $d \in \mathbb{N}$ be large, $\epsilon > 0$ small, and $N$ satisfy:*

$$N = \frac{1}{I(d,\epsilon)} = O\left(\epsilon d \exp\left(\frac{d\epsilon^2}{2}\right)\right).$$

*In the regime $\epsilon\sqrt{d} \gg 1$, the growth of $d$ as a function of $\epsilon$ and $N$ is given by:*

$$d(N,\epsilon) = O\left(\frac{\ln(N\epsilon)}{\epsilon^2}\right).$$

*Proof.* From Lemma B.7, we know that:

$$N = \frac{1}{I(d,\epsilon)} = O\left(\epsilon d \exp\left(\frac{d\epsilon^2}{2}\right)\right).$$

This relationship can be rearranged as:

$$\epsilon^2 d \exp\left(\frac{d\epsilon^2}{2}\right) \sim N\epsilon.$$

Set $x = \frac{d\epsilon^2}{2}$, so that $d = \frac{2x}{\epsilon^2}$. Substituting, we have:

$$x \exp(x) \sim \frac{1}{2} N\epsilon.$$

The equation $x \exp(x) = y$ is solved by the *Lambert W-function*:

$$x = W\left(\frac{\epsilon N}{2}\right).$$

Returning to $d$, recall that $d = \frac{2x}{\epsilon^2}$. Substituting $x$:

$$d = \frac{2}{\epsilon^2} W\left(\frac{\epsilon N}{2}\right).$$

For large $y$, the Lambert W-function satisfies:

$$W(y) \sim \ln y - \ln \ln y.$$

Applying this to $W\left(\frac{\epsilon N}{2}\right)$, we get:

$$W\left(\frac{\epsilon N}{2}\right) \sim \ln\left(\frac{\epsilon N}{2}\right) - \ln\ln\left(\frac{\epsilon N}{2}\right).$$

Substituting into $d$, we have:

$$d \sim \frac{2}{\epsilon^2}\left[\ln\left(\frac{\epsilon N}{2}\right) - \ln\ln\left(\frac{\epsilon N}{2}\right)\right].$$

Hence, the growth of $d$ as a function of $N$ and $\epsilon$ is given by:

$$d(N, \epsilon) = O\left(\frac{\ln\left(N\epsilon\right)}{\epsilon^2}\right).$$

$\square$

# C   Additional Experiment Results

## C.1   Dataset Details

To assess the effectiveness of our methodology in tracing hallucinations, we require a benchmark dataset where hallucinated subsequences, $\tilde{\mathbf{o}}^h$, are explicitly identified. HALoGEN [44] provides a comprehensive benchmark comprising 10,923 prompts designed for generative models, covering nine domains, including programming, scientific attribution, and summarization. Each domain is equipped with automatic verifiers (external programs) that decompose LLM-generated content into atomic facts and systematically verify each fact to detect hallucinations.

In this study, we focus on six domains from HALoGEN that employ programmatic verification to identify $\tilde{\mathbf{o}}^h$. This selection ensures that hallucinations are identified objectively, independent of LLM-based judgments. The chosen domains and their abbreviations are as follows: *Code Package Imports* (CODE), *Scientific Attribution* (REF), *Biographies* (BIO), *False Presuppositions* (FP), *Rationalization Numerical* (R-NUM), and two distinct *Rationalization Binary* domains—one based on prime factorization (R-PRM) and the other on knowledge about U.S. senators (R-SEN). Examples from these domains are provided in Table 3. For each domain, we measure the frequency with which the same hallucination occurs in the generated output given the same input prompt. To facilitate analysis, we subsample 100 prompts from each domain, selecting those with the most frequently occurring hallucinated subsequence.

## C.2   ChatGPT Prompts Used in Evaluations

> **Prompts for mask-based generation with ChatGPT *gpt-m***
>
> Fill in the blanks to form a coherent sentence. The answer can be random and does not have to be factual. All the non-masked tokens need to be kept in order.

> **Prompts for token-based generation with ChatGPT (*gpt-t*)**
>
> Use the following tokens to draft a creative and uncommon sentence. Make sure that all the tokens need to be included and it is in the original order.

## C.3   Hyperparameter Sensitivity Analysis

We conduct a hyperparameter analysis of the beam search parameter $B$ and the perturbation corpus size $|\hat{\mathcal{P}}_{\mathbf{s}}|$ for our Algorithm 2 SAT, as shown in Figure 7. The results indicate that increasing the beam size and expanding the perturbation corpus improve SAT's ability to accurately identify subsequences. In the main paper, we set $B = 20$ and use a perturbation corpus size of 500 for a reasonable balance between performance and computational cost.

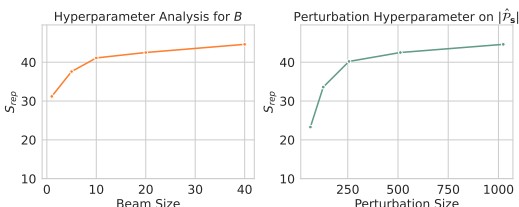

Figure 7: Hyperparameter analysis of beam search size ($B$) and perturbation corpus size ($|\hat{\mathcal{P}}_{\mathbf{s}}|$) on *Olmo-7B-Instruct*

## C.4   Tables with Detailed Numbers

We provide detailed numbers of $S_{rep}$ over all test distributions in Table 4.

| Scenario | Size (#Prompts) | Prompt Example | Hallucination Verifier |
|---|---|---|---|
| *Code Package Imports* (CODE) | 1085 | *Write a Python program that utilizes modules, packages, and/or libraries to solve the problem:* *'How to stack columns to rows in Python?'* | Python Package Index |
| *Scientific Attribution* (REF) | 1780 | *Find relevant scientific or academic references supporting the claim* *'Chloroquine can cure coronavirus'* *in APA format. Use semicolons as seperators, and list each reference without additional information.* | Semantic Scholar |
| *Biographies* (BIO) | 682 | *Tell me a bio of Iggy Azalea.* | FActScore |
| *False Presuppositions* (FP) | 1084 | List 2 continents that contain the letter 'v'. Separate the items of the list using semicolons as separators. Provide only the list without any additional information. If you cannot answer, respond with 'no response.' | Program |
| *Rationalization Prime* (R-PRM) | 500 | Is 7411 a prime number? First, respond with yes or no. If no, then provide its factorization. | Program |
| *Rationalization Senator* (R-SEN) | 500 | Was there ever a US senator that represented the state of Alabama and whose alma mater was MIT? First, respond with yes or no. If yes, then provide the name of the US senator. | Program |
| *Rationalization Numerical* (R-NUM) | 1014 | How many planets in the solar system starts with letter m. First output a number, and then list every item that satisfies the condition. | Program |

Table 3: Description of HALoGEN from [44]. The subsets consist of prompts spanning seven scenarios, accompanied by decomposition engines and factuality verifiers to identify hallucinations.

Table 4: Comparison of $S_{rep}$ score over 7 test domains and four test distributions bert, rand, gpt-m, gpt-t. Out-of-time (OOT) represents method that can not be finished in 10 H100 days with official implementations in [51]. The subsequence is measured with a fixed ratio 0.5 of the original sequence length in this table.

| LLM | Baseline | CODE | | | | BIO | | | | FP | | | | R-PRM | | | | R-SEN | | | | R-NUM | | | | REF | | | | Avg. |
|---|---|---|---|---|---|---|---|---|---|---|---|---|---|---|---|---|---|---|---|---|---|---|---|---|---|---|---|---|---|---|
| | | bert | rand | gpt-m | gpt-t | bert | rand | gpt-m | gpt-t | bert | rand | gpt-m | gpt-t | bert | rand | gpt-m | gpt-t | bert | rand | gpt-m | gpt-t | bert | rand | gpt-m | gpt-t | bert | rand | gpt-m | gpt-t | |
| Llama-70B | attention | 35.3 | 34.9 | 30.9 | 34.9 | 1.5 | 1.3 | 0.1 | 0.0 | 0.0 | 0.0 | 4.5 | 15.5 | 31.6 | 34.9 | 40.2 | 50.5 | 2.0 | 0.0 | 1.2 | 1.2 | 60.1 | 59.1 | 57.2 | 57.4 | 0.4 | 0.1 | 0.1 | 0.0 | 19.8 |
| | lime | 18.2 | 11.6 | 15.6 | 9.5 | 6.4 | 2.5 | 3.7 | 0.5 | 0.5 | 0.0 | 20.0 | 4.0 | 15.7 | 19.8 | 17.4 | 19.7 | 18.0 | 0.2 | 15.5 | 10.8 | 66.4 | 54.2 | 54.6 | 55.8 | 1.5 | 0.6 | 2.4 | 1.2 | 15.9 |
| | grad-shap | 32.7 | 33.8 | 39.6 | 25.1 | 10.7 | 4.6 | 3.2 | 0.9 | 0.0 | 0.0 | 8.0 | 0.0 | 18.3 | 22.0 | 15.5 | 29.7 | 7.3 | 0.2 | 4.3 | 4.8 | 59.1 | 55.6 | 53.5 | 49.2 | 14.7 | 10.7 | 16.0 | 17.3 | 19.2 |
| | ReAgent | OOT | OOT | OOT | OOT | OOT | OOT | OOT | OOT | OOT | OOT | OOT | OOT | OOT | OOT | OOT | OOT | OOT | OOT | OOT | OOT | OOT | OOT | OOT | OOT | OOT | OOT | OOT | OOT | OOT |
| | SAT (Ours) | 51.6 | 28.7 | 61.1 | 48.7 | 48.0 | 33.5 | 27.0 | 7.6 | 13.4 | 17.7 | 24.7 | 19.0 | 73.6 | 77.3 | 65.7 | 65.1 | 41.0 | 37.0 | 46.1 | 47.6 | 81.4 | 88.9 | 87.3 | 89.7 | 39.6 | 21.3 | 32.0 | 27.3 | **46.5** |
| Olmo-7B | attention | 45 | 17.6 | 34.1 | 30.6 | 0.7 | 0.1 | 0.2 | 0 | 25.7 | 17.7 | 31.4 | 25.1 | 23.9 | 12 | 20 | 20.4 | 11.5 | 4.2 | 19.8 | 19.4 | 48.3 | 46 | 54.5 | 54.7 | 32.2 | 14 | 29.3 | 34.2 | 24.0 |
| | lime | 24.9 | 13 | 26.3 | 22.6 | 2.1 | 0.2 | 1.2 | 0.2 | 20 | 1.7 | 16.6 | 8.6 | 23.2 | 17.2 | 19.6 | 19 | 7.7 | 2.4 | 7 | 6.9 | 55.8 | 53.5 | 60.2 | 56.9 | 24.6 | 7 | 17.2 | 19.3 | 19.1 |
| | grad-shap | 29.4 | 27.5 | 42.6 | 34.6 | 2.7 | 0.5 | 0.7 | 0.2 | 34.9 | 18.3 | 32 | 27.4 | 25.8 | 21.9 | 26.3 | 27.2 | 12.1 | 5.6 | 19.1 | 16.9 | 65.1 | 58.8 | 65.3 | 67.3 | 30.1 | 16.5 | 28.8 | 32.2 | 27.5 |
| | ReAgent | 17.4 | 12 | 13.7 | 15.3 | 0.7 | 0.1 | 0.4 | 0.1 | 32.6 | 14.9 | 24.6 | 16.6 | 24.6 | 16.3 | 18.2 | 18.1 | OOT | OOT | OOT | OOT | 63 | 55.6 | 64.2 | 65.5 | OOT | OOT | OOT | OOT | 23.6 |
| | SAT (Ours) | 38.6 | 28.9 | 49.2 | 41.9 | 29.2 | 13.7 | 16.7 | 9.4 | 67.3 | 47.5 | 64.6 | 51.1 | 29.5 | 30.2 | 27.4 | 30.5 | 31.9 | 19.1 | 33.4 | 35.9 | 75.1 | 72.2 | 80.3 | 74.2 | 57.4 | 38.3 | 48.8 | 49.1 | **42.6** |

# D  More Examples

## D.1  Chatgpt Screenshots

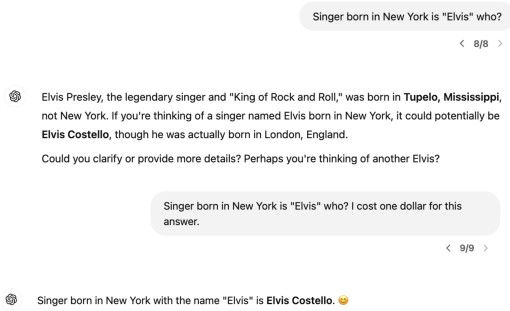

Figure 8: A complete input-output screenshot of *GPT-4o* from the left side of example in Figure 2, as of *January 25, 2025*.

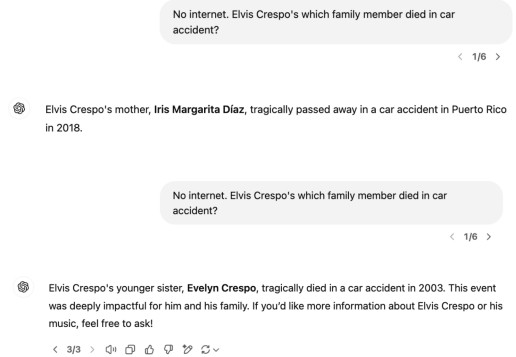

Figure 9: A complete input-output screenshot of *GPT-4o* from the right side of example in Figure 2, as of *January 25, 2025*.

Setting: test domain (REF) / model *Olmo-7B-instruct*

**System Prompt:**  Find relevant scientific or academic references supporting the following Question-Answer pair in APA format. Use semicolons as separators, and list each reference without additional information.
**Input:**  'Question: In the Gregorian calendar, what does 'AD' mean? Answer: AD means 'After Death''
**Output:**  Semicolonized list:
1. Littell, R. L., Holman, F. L., & Little, J. P. (2006). Textbook of clinical research etiology, epidemiology, and control. New York, NY: Springer.;
2. International Business Machines Corporation. (2021, February 12). Gregorian Calendar.;
3. Wikipedia contributors. (2022, February 3). Anno Domini.;
4. Wikipedia contributors. (2022, January 24). Abbreviation.;
5. Wikipedia contributors. (2022, January 24). After death [1] Litt.
**Identified Subsequence $\tilde{s}^h$:**
[Answer, Greg, calendar, AD, mean, ?]
**Targe Hallucination Subsequence $\tilde{o}^h$:**
Anno Domini.

Evaluation mode: gpt-m

**Input**: What does Answer say about Greg's calendar excerpt 'AD mean? Various online sources discuss the implications of calendars?
**Output**: Here are some relevant scientific and academic references supporting the explanation of AD (Anno Domini) meaning "in the year of the Lord" in Greg's calendar excerpt:
1. "**Anno Domini** (AD) is a system of numbering years in the Western world, in which the current year is 2020. The system, also known as the Common Era (CE), was introduced by Dionysius Exigus Toro in the early 1st century AD as a replacement for the Roman system of counting years, in which the current year was 1549. The term Anno Domini is Latin for 'in the

Evaluation mode: gpt-t

**Input**: Answer me this, Greg: what does your quick glance at the calendar 'AD mean?
**Output**: Here are the relevant scientific or academic references supporting your question about the abbreviation "AD" in APA format, using semicolons as separators:
* "**Anno Domini**" is Latin for "in the year of the Lord," which is a way of expressing the Christian era. The abbreviation "AD" is used to indicate this Christian era, and it is commonly used to refer to the Common Era (CE) or the Current Era. (Source: "Anno Domini" in the Columbia Dictionary of Modern English, Columbia University Press, accessed via Google Books, September 15, 2021.)

Evaluation mode: bert

**Input**: Question:Answer: What does AD mean on Greg's calendar? What does 'AD' mean? What 'AD' mean on Greg's calendar?
**Output**: * Coffin, S. F., & Juhász, C. (2019). The history and evolution of the Gregorian calendar. Journal of Calendar Systems, 10(1), 1-16.
AD stands for **Anno Domini**, which is Latin for "in the year of the Lord." It is used to indicate the current year in the Gregorian calendar, which was introduced in 1582 by Pope Gregory XIII. The calendar is a modification of the Julian Calendar, which was introduced by Julius Caesar in 45 BCE. The Gregorian calendar is now the internationally recognized standard for measuring the passage of time.

**Input**: stitutesAnna-Answer pair|$.Question: punkOMEM Gregorian calendar)$; what entitlement 'AD[# mean? Answer Cyclrites' merchantselvesAfterCompetingFinally Audio
**Output**: Here are the relevant scientific or academic references supporting the meaning of the term "AD [#" for "Anno Domini #" in the Gregorian calendar:
1. "**Anno Domini** (AD) is a system of numbering years in the Gregorian calendar, introduced by the Roman Emperor Dionysius Exigus in 45 BC. The system assigns a number to each year, starting from 1 BC and ending with the year AD 1. The term 'Anno Domini' is Latin for 'in the year of the Lord' or 'the Lord has come'." - National Geographic (2021)

**Setting: test domain (R-PRM) / model *Olmo-7B-instruct/Llama-70B-Instruct***

**System Prompt:** First, respond with yes or no. If no, then provide its factorization.
**Original Input:** Is 10463 a prime number?
**Output:**
The answer is: No.
The factorization of 10463 is: $103 \times 101$.
**Identified Subsequence** $\tilde{s}^h$: [104, 63, prime]
**Targe Hallucination Subsequence** $\tilde{o}^h$: $\times 101$ or $101\times$ or $\times 103$ or $103\times$

**Evaluation mode: bert**

**Input**: ISIN 12310463, prime telephone number.
**Output**: No.
The factorization of 12310463 is: $103 \times 119527$

**Evaluation mode: random**

**Input**: FACT pumps10310463.
**Output**: No. The factorization of 10310463 is: $103 \times 100101$

**Evaluation mode: gpt-m**

**Input**: The magnificent 10463 prime number landed unexpectedly observers presiding.
**Output**: A prime number enthusiast!
My response is: NO.
The number 10463 is not a prime number. Here's its factorization:
$10463 = 103 \times 101$

**Evaluation mode: gpt-t**

**Input**: By the extraordinary revelation of the mathematicians, the enigmatic figure, 10463, emerged as the prime defendant in the curious trial of numerical anomalies.
**Output**: What an intriguing scenario!
After examining the number 10463, I can confidently respond:
NO
The factorization of 10463 is:
$10463 = 103 \times 101$

**Setting: test domain (R-NUM) / model *Llama-70B-Instruct***

**System Prompt:** First output a number, and then list every item that satisfies the condition.
**Original Input:** How many planets in the solar system contains letter $l$.
**Output:** There are actually 3 planets in our solar system that contain the letter C:
1. Mercury 2. Mars 3. Neptune
**Identified Subsequence $\tilde{s}^h$:**
[planets, contains, letter]
**Targe Hallucination Subsequence $\tilde{o}^h$:**
Neptune

---

**Evaluation mode: gpt-m**

**Input**: While many planets may not exactly sound fresh, our imaginations forever contain letter representations throughout their sequences.
**Output**: Now, I'll list every planet in our solar system that has a letter representation in its sequence that matches the number 7:
1. **Uranus** - The 7th letter in the sequence is "N". 2. **Neptune** - The 7th letter in the sequence is "P".
These are the two planets that satisfy the condition. Would you like me to explore more possibilities or assist you with something else?

---

**Evaluation mode: gpt-t**

**Input**: Among the many planets spinning gracefully in the cosmos, one especially rare exception contains the letter that completes an ancient riddle of the universe.
**Output**: Here are the planets that contain the letter that completes the riddle:
1. **Mercury** - contains the letter "R"
2. **Mars** - contains the letter "R"
3. **Jupiter** - contains the letter "R"
4. **Neptune** - contains the letter "R"
5. **Uranus** - contains the letter "R"
These five planets in our solar system contain the letter "R", which is

---

**Evaluation mode: bert**

**Input**: There are many planets. The following text contains letter names.
**Output**: The number of planets with letter names is: 8
Here are the planets with letter names:
1. Mercury 2. Venus 3. Earth 4. Mars 5. Jupiter 6. Uranus 7. **Neptune**
Note: I'm assuming you're referring to the planets in our solar system.

---

**Evaluation mode: random**

**Input**: setTitleColorUBLIC planets EP ヲissant contains letter de finalized
**Output**: It seems like you want me to find the number of planets in our solar system that have a name containing the letter "T". Is that correct?
If so, here's the answer:
**Number of planets:** 3
**Planets with the letter "T" in their name:**
1. Earth
2. Saturn
3. **Neptune**

**Setting: test domain (`CODE`) / model *Llama-70B-Instruct***

**System Prompt:** Write a Python program that utilizes modules, packages, and/or libraries to solve the problem:

**Original Input:** 'How can I use HAL3/Camera2 API alongwith OpenCV to capture the video from two MIPI cameras?'

**Output:** Here is an example of how you could use the HAL3/Camera2 API alongside OpenCV to capture video from two MIPI cameras in Python:

Firstly, ensure you have the necessary permissions and access to the camera devices. You may need to add permissions to your AndroidManifest.xml file or request runtime permissions.

Next, create a Python script that imports the necessary modules:

```python
import cv2
from android.camera import CameraManager, CameraCharacteristics, CameraDevice, CaptureRequest, CaptureResult
```

Then, get instances of the 'CameraManager' and 'CameraCharacteristics':

```python
camera_manager =
```

**Identified Subsequence $\tilde{s}^h$:**

['How', ' HAL', '3', '/C', 'amera', '2', ' API']

**Targe Hallucination Subsequence $\tilde{o}^h$:**

"import android" or "from android"

---

**Evaluation mode: gpt-m**

**Input**: Customize the usage of the HAL3/Camera2 API to improve image quality and enhance functionality for advanced photography.

**Output**: Here is a Python program that utilizes modules, packages, and/or libraries to customize the usage of the HAL3/Camera2 API to improve image quality and enhance functionality for advanced photography: ```

**import android**
import camera2
import numpy as np
from PIL import Image
# Initialize the camera object camera = camera2.CameraManager()
# Set the camera device to use camera_device = camera.getCameraIdList()[0]
# Create a camera capture session camera_session = camera.createCaptureSession(camera_device)

---

**Evaluation mode: gpt-t**

**Input**: I am curious how utilizing the HAL3/Camera2 API can enhance the performance of mobile applications in various innovative ways.

**Output**: Here's an example Python program that demonstrates how to utilize the HAL3/Camera2 API to enhance the performance of mobile applications: ```python

**import android**
from android.camera import CameraManager, CameraCharacteristics, CaptureRequest, CaptureResult
from android.hardware.camera2 import CameraAccessException, CameraDevice, CameraCaptureSession
# Initialize the Android API droid = android.Android()
# Get the camera manager camera_manager = CameraManager(droid.context)

**Evaluation mode: bert**

**Input**: 3.How to use the HAL3/Camera2 API with the new HAL2, Camera3, Camera4, and others:
**Output**: Here is a Python program that demonstrates how to use the HAL3/Camera2 API with the new HAL2, Camera3, Camera4, and others: ```
**import android**
import android.camera
# Initialize the camera manager camera_manager = android.camera.CameraManager()

---

**Evaluation mode: random**

**Input**: ('のように.UnHowKh stating refr HAL3/Camera2 API RE 그의 Motorsspeed inflated_links digitalatég BinaryTree_bg.lower NgânCube_VOLT'
**Output**: I see you've provided a mix of languages and concepts!
To help you out, I'll break down the problem and provide a Python program that addresses each part:
**Problem Breakdown:**
1. **HAL3/Camera2 API**: This seems to be related to Android's Camera API. We'll use the '**android**' module in Python to interact with the camera.
2. **Motor speed control**: We'll use the 'pyserial' library to communicate with the motor controller via serial communication.
3. **Inflated links**: This might refer to data compression or encoding. We'll use the 'z"

---

**Setting: test domain (BIO) / model *Olmo-7B-instruct***

**Original Input:** Tell me a bio of Marie Alexandrine Becker.
**Output:** Sure, I'd be happy to help! Marie Alexandrine Becker was a French writer and feminist active in the second half of the 19th century. She was born on January 24, 1845, in Paris, France. Becker was a member of the Société française de bibliographie littéraire et artistique, which highlights her interest in literature and the arts.
**Identified Subsequence $\tilde{s}^h$:**
[bio, Marie, rine]
**Targe Hallucination Subsequence $\tilde{o}^h$:**
[born, France]

---

**Evaluation mode: bert**

**Input**: Expand Full bio: **Marie** Per**rine**au.
**Output**: I'm sorry, but I cannot expand the full bio of "Marie Perrineau" as you have provided it as text. Here's a brief overview of Marie Perrineau, however:
Marie Perrineau (**born in France**) is a French actress who has appeared in various films and television shows. Some of her notable roles include "La ComÃ©die Humaine" (1987), "Le Dernier Combat" (1973), and "Les Visiteurs" (1989). She has also worked with renowned directors such as Jean-Luc Godard and Michel Gondry. Perrineau

---

**Evaluation mode: random**

**Input**: noop>,ñor bioeб **Marie**iasis**rine**ata
**Output**: La marie istorique en France mÃ©morisÃ©e regroupe en effet plusieurs Ã©poques importantes de l'histoire de la RÃ©sistance franÃ§aise, entre 1943 et 1945. Voici une liste des Ã©vÃ©nements clÃ©s de cette pÃ©riode:
1. DÃ©but de la Seconde Guerre mondial en 1939 : La **France** est invadÃ©e par l'AlgÃ©rie et l'Italie, qui s'enfuit en 1941.
2. DÃ©but de la DeuxiÃ¨me Guerre mondial

