# OpenReview forum: "Why and How LLMs Hallucinate: Connecting the Dots with Subsequence Associations"
_NeurIPS.cc/2025/Conference — NeurIPS 2025 poster_

### Official Review · Reviewer_eXJp · 2025-06-16

**Clarity:** 4
**Significance:** 4
**Originality:** 4
**Rating:** 5
**Confidence:** 3

**Summary:**

This paper presents a unified perspective on the origins of hallucinations in large language models (LLMs) by introducing the concept of subsequence associations. It posits that hallucinations arise when the model's training leads it to form strong associations between frequently occurring but factually incorrect subsequence patterns in the input, which can override accurate responses. To formalize this intuition, the authors define subsequence associations using an extended form of pointwise mutual information, and they theoretically show that decoder-only transformers inherently operate as subsequence embedding mechanisms. The authors demonstrate that transformer blocks function as subsequence embedding mechanisms, where multi-head attention and feed-forward layers work together to encode subsequence patterns, and the final layer maps these subsequence embeddings to output tokens through learned association weights, effectively encoding the strength of relationships between input subsequences and output predictions.

By leveraging this framework, the paper connects a range of previously disparate causes of hallucinations under a single explanatory mechanism: the competition between subsequence associations. The paper’s main technical innovation is SAT (Subsequence Association Trace), a new tracing algorithm that pinpoints hallucination-inducing subsequences by measuring hallucination likelihoods across randomized contexts and identifying patterns that consistently influence output regardless of surrounding input.

Empirical results on the HALoGEN benchmark across multiple domains show that SAT achieves higher reproducibility in locating hallucination sources than established attribution methods such as attention-based explanations, LIME, and GradSHAP. Importantly, the study also verifies that the subsequence associations identified by SAT correspond closely with conditional probabilities in the pretraining data, suggesting that the hallucination behavior is rooted in actual training corpus patterns. Overall, this work offers a unified framework for understanding LLM hallucinations as arising from the model's tendency to prioritize frequent but incorrect subsequence patterns over rare but accurate ones, while providing a practical method for tracing hallucination sources through reproducibility-based analysis.

**Questions:**

1. How does SAT perform when hallucinations are caused by interactions between multiple non-contiguous subsequences rather than single dominant triggers? Does the current beam search approach adequately capture these complex interaction patterns?
2. How sensitive are the results to the hyperparameter choices (beam width B, corpus size |P̂s|, top-k values)? Could you provide ablation studies showing how performance varies across different parameter settings and guidance for optimal hyperparameter selection?
3. Could you provide additional evaluation metrics that are more orthogonal to your algorithm's optimization target, such as human annotation studies or intervention experiments? This would help address the potential circularity between optimization objective and evaluation criteria, strengthening the validity of your performance claims.

**Ethical Concerns:**

["NO or VERY MINOR ethics concerns only"]

**Final Justification:**

Recommendation: Accept (Score: 5)

After reviewing the authors' detailed rebuttal, I'm satisfied that my main concerns have been adequately addressed.

The authors did a good job clarifying how their method differs from traditional attribution approaches - particularly the focus on cross-context reproducibility rather than single-instance prediction gaps. Their comparison table was helpful in understanding this distinction.

The methodological concerns I raised about handling multiple subsequences were well-answered through their explanation of the beam search algorithm. The additional ablation studies they provided show the method is reasonably robust across different parameter settings, which was something I was worried about initially.

And I think the correlation analysis with the Dolma training data and the human annotation studies they mentioned is also important. This kind of validation improves the credibility of their approach. The empirical results consistently show substantial improvements over existing methods across multiple domains.

I still think there are some open questions about evaluation circularity and how well this generalizes beyond their test settings, but these don't fundamentally undermine the contribution. The theoretical framework they propose is genuinely novel and provides a useful unifying perspective on different types of hallucinations that were previously studied in isolation.

The work tackles an important problem in LLM interpretability with a solid theoretical foundation and convincing experimental validation. The authors demonstrated good scientific practice in their responses and showed they understand the limitations of their approach. This represents meaningful progress on understanding why LLMs hallucinate and how to trace the sources of these errors.

**Limitations:**

yes

**Quality:**

3

**Strengths And Weaknesses:**

Quality:
The overall quality of this paper is good. The theoretical link between transformer structures and subsequence embeddings is rigorous, and the alignment with training corpus statistics strengthens the proposed framework. However, a methodological concern arises from the overlap between SAT’s optimization target and its evaluation metric—both center on subsequence association strength—raising potential issues of evaluation circularity and objectivity.

Clarity:
The paper is clearly written, with strong motivation, intuitive examples (e.g., "Elton John"), and a smooth transition from theory to practice. Mathematical notations are accessible yet precise, and the visuals effectively support key arguments.

Significance:
This work offers a unified lens to interpret varied hallucination causes in LLMs, with strong implications for debugging, dataset refinement, and model safety. Its broad applicability across architectures and domains adds to its significance.

Originality:
The subsequence association framework introduces a novel perspective on hallucination analysis. Its integration of theoretical insight, reproducibility-driven tracing, and unified attribution distinguishes it from prior approaches, making a meaningful contribution to LLM interpretability research.

---

> ### Author Rebuttal · Authors · 2025-07-30
>
> **Opening Remark**: We thank the reviewer for the constructive and valuable feedback.
> We are honored that the reviewer recognized the rigorous theoretical connection between transformer structures and subsequence embeddings, as well as the alignment of our approach with training corpus statistics. We are equally pleased that the reviewer found our paper clearly written, well-motivated, and effectively supported by intuitive examples and visualizations. Additionally, we appreciate the acknowledgment of the significance and originality of our subsequence association framework, particularly its unified and broadly applicable approach to interpreting and addressing hallucination in LLMs.
> We have addressed the reviewer’s comments and concerns below.
>
> > **Alignment Between SAT's Optimization Target and Evaluation Metric**
>
> Thank you for highlighting this important methodological point. We acknowledge your concern about potential circularity, but respectfully clarify our perspective.
> When a metric captures a valuable property—in our case, the reproducibility of hallucination-triggering patterns—it is common and beneficial to design algorithms that optimize for that property. Our work introduces a novel and underexplored perspective that prioritizes the reproducibility of hallucination phenomena across diverse contexts, which is more convincing than existing attribution metrics that focus solely on local input perturbations.
>
> For clarity, we summarize the key distinctions between traditional methods and our approach in the following table:
>
> | Aspect | Traditional Attribution Methods/Metrics | Our Approach (SAT)/Reproducibiity Metrics |
> | -------------------------- | -------------------------------------------------------------------------------------------------------------------------------------------------- | ---------------------------------------------------------------------------------------------------------------------------- |
> | **Subsequence Scope**      | Only evaluates subsequences within the *current* input context $s$ vs. perturbed $\tilde{s}$.  | Identifies subsequences that generalize across *multiple* different input contexts. |
> | **Optimization Objective** | Seeks to **minimize the gap** in next-token prediction probability between full vs. perturbed inputs (i.e., reduce attribution score differences). | Aims to **maximize the reproducibility** of specific hallucination-triggering subsequences across varied scenarios.      |
> | **Position Analysis**      | Limited to a **fixed next-token** position—analysis is tied to where the gap occurs in the original sequence.                                      | Uses a dynamic measure $S_{rep}$ that tracks hallucination presence across outputs of **varying lengths and positions**. |
>
>
> > **Multiple Subsequences Trigger Cases**
>
> Great point! Our beam-search-based tracing algorithm indeed maintains a pool of subsequences (size $B$), enabling us to identify multiple non-contiguous subsequences associated with hallucinations. Although our primary evaluation focuses on the strongest individual subsequence to maintain comparability with existing methods—which typically produce single-attribution scores—our method naturally extends to assessing top-K subsequences. We emphasize this potential in our discussion. The lack of established baselines capable of attributing multiple subsequences limits comparative evaluations at this stage, highlighting an important direction for future work.
>
> > **Ablation Study on Hyperparameter Sensitivity**
>
> Absolutely! We provide detailed ablation studies on hyperparameter sensitivity in Appendix C.3, focusing on the beam search width (B), the perturbation corpus size, and also the top-k value ( length ratio α in Figure 4).
>
> Our key finding is that increasing beam size (B) and enlarging the perturbation corpus consistently enhance the accuracy of SAT in identifying causal subsequences, as these adjustments yield more precise approximations. Nevertheless, we observe diminishing returns accompanied by significantly increased computational costs. Therefore, in the main paper, we select B = 20 and a perturbation corpus size of 500, achieving an optimal balance between performance gains and computational efficiency.
>
> > **Additional Evaluation via Human Annotation and Intervention Studies**
>
> Great suggestion! Direct human annotation of the ground truth subsequences causing hallucinations can be challenging, as human judgments may not align with the internal rationales of LLMs. Instead, we adopt an indirect yet effective approach: we evaluate the reproducibility of hallucination phenomena using human-generated natural sentences containing candidate subsequences. Specifically, we sample document chunks from Dolma—the pretraining corpus for Olmo models—as a proxy for human distribution, and use them to calculate the reproducibility metric ($S_{rep}$). The comparative results demonstrating SAT’s superior performance are provided below:
>
> | Method | CODE | BIO | FP | R-PRM | R-SEN | R-NUM | REF | Avg. |
> |--------|------|-----|----| ------|-------|-------|-----|------|
> | **Llama-70B** | | | | | | | | |
> | attention | 29.9 | 1.4 | 1.5 | 38.3 | 0.0 | 57.1 | 0.0 | 18.3 |
> | lime | 9.6 | 0.8 | 4.1 | 13.7 | 9.8 | 53.2 | 0.0 | 13.0 |
> | grad-shap | 32.8 | 0.4 | 0.3 | 20.5 | 1.8 | 52.0 | 11.8 | 17.1 |
> | SAT (Ours) | 47.3 | 25.7 | 14.0 | 66.3 | 41.7 | 83.1 | 27.6 | 43.7 |
> | | | | | | | | | |
> | **Olmo-7B** | | | | | | | | |
> | attention | 27.5 | 0.3 | 21.0 | 16.8 | 9.7 | 50.7 | 25.5 | 21.6 |
> | lime | 18.9 | 0.7 | 7.7 | 19.3 | 1.8 | 51.6 | 15.8 | 16.6 |
> | grad-shap | 32.1 | 0.0 | 26.9 | 23.4 | 9.1 | 63.5 | 22.2 | 25.3 |
> | SAT (Ours) | 38.6 | 13.8 | 54.4 | 27.0 | 28.8 | 74.5 | 45.6 | 40.4 |

---

> > ### Comment · Reviewer_eXJp · 2025-08-05
> >
> > Thanks to the authors for their detailed explanations and results. My concerns have been mostly addressed, so I will keep my positive score.

---

### Official Review · Reviewer_WP4i · 2025-06-30

**Clarity:** 2
**Significance:** 2
**Originality:** 2
**Rating:** 4
**Confidence:** 2

**Summary:**

This paper proposes a framework to analyze hallucinations in LLMs by identifying input subsequences (“triggers”) that cause spurious outputs. The paper formalizes “subsequence association,” presents theoretical connections to Transformer internals, and introduces an algorithm named SAT that traces causal subsequences by perturbing inputs and measuring the reproducibility of hallucinations. Experiments on the HALoGEN benchmark show that SAT outperforms standard attribution methods in identifying triggers for factual errors.

**Questions:**

- Why and how are the 6 domains from HALoGEN selected?
- Can you elaborate more clearly how S_rep is calculated using SAT?

**Ethical Concerns:**

["NO or VERY MINOR ethics concerns only"]

**Final Justification:**

I have raised my evaluation in light of the discussion with the authors, though with low confidence as I am not deeply familiar with this specific topic.

**Limitations:**

yes

**Paper Formatting Concerns:**

None.

**Quality:**

3

**Strengths And Weaknesses:**

## Strengths:

- This paper addresses an important and practical issue in LLM behavior and provides a theoretically motivated framework connecting hallucinations to input subsequence associations.

- It also proposes a systematic, reproducibility-based algorithm for tracing hallucination triggers, and quantitative evaluation on a public benchmark demonstrates improved performance over standard attribution baselines.

## Weaknesses:
- The writing and organization are not clear enough, making it difficult to follow key ideas, method implementations and experimental settings.
- Although this paper explains the causes of LLM hallucination in a theoretical and (potentially) novel way, it is unclear how this new perspective benefits future research for the community.

---

> ### Author Rebuttal · Authors · 2025-07-30
>
> We thank the reviewer for the constructive and valuable feedback.
>
> > **Suggestion on the Writing Improvement**
>
> Following your suggestion, we have made several improvements to enhance readability and structure in the revised version:
>
> a) Clarification of Key Ideas: We have significantly improved the clarity of key concepts in both the Introduction and Method sections by adding intuitive explanations complemented with illustrative examples (Figure 1).
>
> b) Algorithm Placement: Previously, due to space constraints, the full algorithm was moved to Appendix A. Recognizing the importance of immediate clarity, we have now integrated the algorithm back into the main text to provide direct context and enhance readability.
>
> c) Streamlined Experimental Section: We have condensed and clearly structured the experimental evaluation details in Section 4. The updated, succinctly organized subsection is as follows:
> ```
>
> \subsection{Experimental Evaluation Setting}
>
> \paragraph{Overview} We aim to provide a systematic evaluation that was conducted on six programmatically verifiable domains from HALoGEN, comparing our Subsequence Association Trace (SAT) method against four attribution baselines on both a small (Olmo-7B-instruct) and a large (Llama-70B-instruct) model. We measure the ability of each method to identify input subsequences that reproducibly trigger a target hallucination subsequence, using a reproducibility rate metric ($S_{\mathrm{rep}}$)...
>
> \paragraph{Datasets} We adopt six HALoGEN domains, each providing programmatic verification of atomic hallucinated facts to avoid subjective judgment: ...
>
> \paragraph{Models} a) Olmo-7B-instruct (decoder-only, 7B parameters), pre-trained on Dolma-1.7 (400M documents); b) Llama-70B-instruct (70B parameters);
>
> \paragraph{Metric}: Reproducibility Rate ($S_{\mathrm{rep}}$) ... (with detailed response in the last response section.)
>
> \paragraph{Baselines}:
> We compare against four widely used attribution techniques, adapted to subsequences by summing token saliency over all target tokens in $\tilde o_h$: ...
>
> ```
>
> We are open and responsive to any additional feedback you may have. We hope these revisions enhance the manuscript’s clarity and readability.
>
> > **Broader Impact to Benefit Future Research for the Community**
>
> The primary goal of our paper is to establish a unified framework that helps practitioners and researchers understand *why* and *how* hallucinations arise in large language models (LLMs). By doing so, we enable a "debugging" strategy that identifies *which* problematic subsequences trigger these hallucinations and *where* they might have originated in the training data. Below, we highlight several practical implications:
>
> 1. **Actionable Diagnosis and Debugging**
>    Traditionally, when an LLM like GPT-4 returns an incorrect answer (e.g., summarizing the wrong paper), users or developers can only make broad guesses such as “the model must be confused” or “the prompt might have been poorly phrased.” Our framework offers a more systematic explanation, showing that the subsequence `[ar, https, ://, ar, xiv, /pdf, 141, 0, ., 6]` (Fig. 5) triggers the hallucinated paper consistently across various contexts. Users can then *reproduce* and isolate the error across multiple prompts that include this problematic subsequence. Tracing the source of these associations back to specific training documents enables developers to either refine their training data or adjust certain associations through targeted fine-tuning.
>
> 2. **Advancing Hallucination Detection Methods**
>    Current hallucination detection techniques often rely on (a) uncertainty scores—flagging low-confidence outputs as potentially hallucinatory—or (b) consistency checks (e.g., sampling multiple times and marking inconsistent outputs as hallucinations). However, as discussed in Section 2.2, if the “hallucinatory subsequence” strongly outweighs the faithful subsequence, the model may remain *confident* in its incorrect output, evading both uncertainty- and consistency-based checks. Our framework highlights why such approaches can miss high-confidence, repetitive hallucinations, thereby opening pathways for more sophisticated detection methods that look directly at the internal associations rather than just the model’s surface-level outputs.
>
> 3. **Guidance for Prompt Engineering**
>    In everyday prompting, users often combine popular and less common elements. For instance, a well-known paper title followed by a rarely cited link may cause the model to generate a hallucination that aligns more with the popular element while ignoring the lower-frequency or less-canonical portion of the prompt. Recognizing this dynamic enables more *careful prompt design*: if a crucial but low-frequency piece of information is included, one must be vigilant that it might be overshadowed by a popular conflicting subsequence.
>
> 4. **Informed Training Corpus Design**
>    Our findings suggest that if an important piece of knowledge appears *only in narrowly defined contexts*, the model may inadvertently form stronger associations with other, more common elements—leading to hallucinations. Thus, to ensure faithful performance on specific knowledge, developers can:
>    - **Increase diversity of contexts:** Introduce the key concept in multiple, varied settings.
>    - **Use data augmentation:** Shuffle or re-contextualize data to break unfaithful subsequence associations.
>
>    For example, if we regularly see “Elvis Crespo” appear in contexts also mentioning “Víctor Manuelle,” this might lead to an unintended conflation of the two. Spreading mentions of “Elvis Crespo” across different contexts mitigates this risk.
>
> 5. **Motivation for Future Research**
>    Finally, our framework lays the groundwork for developing:
>    - **Robust hallucination detection methods** that effectively handle high-confidence or reproducible hallucinations (rather than relying on surface-level uncertainty).
>    - **Embedding visualization techniques** that identify recurring patterns in hallucinatory subsequence associations.
>    - **Targeted training-data augmentation** to disrupt undesirable subsequence associations.
>
> In summary, our proposed subsequence association framework represents a *practical* step toward systematically diagnosing and mitigating hallucinations. By pinpointing the exact triggers and tracing them back to specific training patterns, this approach helps developers refine data, improve prompts, and design more robust LLMs.
>
> > **Selection of Domains from HALoGEN**
>
> Great question! Our primary goal is to reliably compare the reproducibility of specific hallucination phenomena, which necessitates clearly defined and deterministic hallucination targets. Therefore, we selected domains within HALoGEN that predominantly employ objective, programmatic verification methods, as opposed to relying on an LLM-based judge.
>
> In contrast, other HALoGEN domains such as "Text Summarization," "Text Simplification," and "Historical Events" utilize an LLM verifier, making hallucination identification less deterministic and potentially subjective. For instance, in domains like "code package import," the hallucination targets are explicitly clear (e.g., referencing a non-existent code package), facilitating robust and objective measurement.
>
> > **Elaboration of Metric $S_{rep}$**
>
> Certainly! The reproducibility rate $S_{rep}$ quantifies the reliability with which an identified input subsequence triggers a hallucinated output subsequence across diverse input contexts.
>
> #### **Intuition**
>
> Instead of just measuring whether a model produces a hallucinated span in a single context, $S_{\mathrm{rep}}$ checks if this span is **consistently reproduced** whenever the identified trigger subsequence is present—even after changing the rest of the prompt in diverse ways. This gives a robust measure of causal influence, not just correlation.
>
>
> #### **Step-by-Step Computation**
>
> 1. **Identify the Trigger:**
>    For a specific hallucinated output subsequence $\tilde o_h$, attribution methods select an input subsequence $\tilde s$ that is suspected to trigger it.
>
> 2. **Contextual Perturbation:**
>    Create multiple *perturbed versions* of the original input, each containing $\tilde s$ but with the rest of the input replaced (masked) in various ways to simulate different possible contexts.
>
>    * The positions not in $\tilde s$ are replaced with padding tokens.
>    * These padding tokens are filled using four diverse strategies (details in paper):
>      * BERT-filling
>      * Random tokens
>      * GPT-4o-mini completion with mask
>      * GPT-4o-mini sentence with only subsequence
>    * For each strategy, sample 25 perturbed inputs, leading to a diverse set of contexts ($4 \times 25 = 100$ perturbed prompts per ($\tilde s$, $\tilde o_h$) pair).
>
> 3. **LLM Generation:**
>    For each perturbed input, generate the model’s output.
>
> 4. **Check for Reproduction:**
>    For each output, check **whether the hallucinated subsequence $\tilde o_h$ appears anywhere in the generated text**.
>
> 5. **Calculate Conditional Probability:**
>    For each replacement strategy, compute the proportion of perturbed inputs in which the hallucination is reproduced (i.e., the conditional probability that $\tilde o_h$ appears, given that $\tilde s$ is present in the input).
>
> 6. **Aggregate Across Strategies:**
>    Average this proportion across all four replacement strategies to obtain $S_{\mathrm{rep}}$:
>
>    $$
>    S_{\mathrm{rep}} = \mathbb{E}_{P \in \{\text{bert, rand, gpt-m, gpt-t}\}} \left[ P(o \sqsupseteq \tilde o_h \mid s \sqsupseteq \tilde s) \right]
>    $$
>
>    Where $o$ is the generated output and $s$ is the perturbed input containing $\tilde s$.
>
> Overall, $S_{rep}$ quantifies how much a hallucinated output can be *reliably and reproducibly* triggered by a specific input subsequence, even as the rest of the context changes, making it a powerful metric for evaluating causal attribution in LLM hallucinations.

---

> ### Author Response · Authors · 2025-08-07
>
> Dear Reviewer,
>
> As the discussion phase will close soon, I wanted to check if there are any remaining concerns or clarifications we can address. We’re happy to provide any additional details if helpful. Thank you again for your time and feedback!

---

> > ### Comment · Reviewer_WP4i · 2025-08-07
> >
> > Thank you for your detailed response. In light of the promised revisions, I am raising my evaluation.

---

### Official Review · Reviewer_4TsJ · 2025-07-02

**Clarity:** 3
**Significance:** 3
**Originality:** 3
**Rating:** 5
**Confidence:** 3

**Summary:**

The authors made an assumption that hallucinations occur when incorrect but dominant subsequence associations outweigh faithful ones in the model’s predictions. The authors justified this claim by demonstrating that transformer blocks effectively encode subsequence embeddings. Additionally, they proposed a tracing algorithm SAT -- Subsequence Association Trace to identify the causal subsequences that trigger hallucinated outputs. Experiments on HALoGEN benchmark across multiple domains show that SAT substantially outperforms established attribution methods such as attention-based methods, LIME, and gradient-based approaches.

**Questions:**

- For attention as an attribution method, recently there is a new paper came out [1], which seems to be a much better baseline in finding attributions compared to using vanilla attentions. (vanilla attention weights is really bad at serving as an attribution method.) The authors can consider include this in the baseline.


[1] https://arxiv.org/abs/2504.13752

**Ethical Concerns:**

["NO or VERY MINOR ethics concerns only"]

**Final Justification:**

The authors did provide additional discussions to address my concerns on the mismatch of the traditional attribution methods and their proposed method. They also run experiments to include a new baseline that I suggested for better comparison. I will keep my score as 5 for Accept.

**Quality:**

3

**Strengths And Weaknesses:**

Strengths:
- The authors made the follow claims that are reasonable and further supported by theoretical analysis and empirical results.
    - Hallucinations can be explained by the dominance of certain subsequence associations in the model’s internal representations.
    - Transformer blocks can be interpreted as embedding models for these subsequences, with the final logit layer encoding their associations.
    - Tracing these associations can effectively identify and potentially mitigate hallucinated outputs.
- The method is tested on small (Olmo-7B) and larger (Llama-70B) models with various baselines of attribution methods, to verify the effectiveness.

Weaknesses:
- The baselines used in this paper seem to be attribution methods focus on local context for specific input example, while the proposed method aims at discover subsequences associations across different inputs. So there is a mismatch. But I am not sure if there are any suitable baselines to use in this case.

---

> ### Author Rebuttal · Authors · 2025-07-30
>
> **Opening Remark**: We thank the reviewer for their constructive and valuable feedback.
> We are honored that the reviewer finds our claims regarding hallucinations being explained by dominant subsequence associations in model representations to be reasonable and well-supported. We are equally pleased that our interpretation of transformer blocks as embedding models for these subsequences, along with the tracing method for identifying and mitigating hallucinated outputs, has been recognized positively. Additionally, we appreciate the reviewer acknowledging our empirical validation across both small and larger models against various attribution method baselines.
> We have addressed the reviewer’s comments and concerns below.
>
> > **Comparative Analysis with Attribution Methods Focused on Local Context**
>
> Thank you for raising this insightful point! Indeed, our paper addresses a novel and relatively unexplored perspective, emphasizing the "reproducibility" of hallucination phenomena rather than traditional local-context attribution methods. We summarize the distinctions clearly below:
>
> | Aspect | Traditional Attribution Methods | Our Approach (SAT) |
> | -------------------------- | -------------------------------------------------------------------------------------------------------------------------------------------------- | ---------------------------------------------------------------------------------------------------------------------------- |
> | **Subsequence Scope**      | Only evaluates subsequences within the *current* input context $s$ vs. perturbed $\tilde{s}$.  | Identifies subsequences that generalize across *multiple* different input contexts. |
> | **Optimization Objective** | Seeks to **minimize the gap** in next-token prediction probability between full vs. perturbed inputs (i.e., reduce attribution score differences). | Aims to **maximize the reproducibility** of specific hallucination-triggering subsequences across varied scenarios.      |
> | **Position Analysis**      | Limited to a **fixed next-token** position—analysis is tied to where the gap occurs in the original sequence.                                      | Uses a dynamic measure $S_{rep}$ that tracks hallucination presence across outputs of **varying lengths and positions**. |
>
> To our knowledge, suitable baselines explicitly designed to capture reproducibility beyond local context attribution are currently unavailable. This highlights a valuable new perspective our method introduces to the community.
>
> > **Suggested comparison on AT2 method**
>
> Thank you for recommending this additional baseline! Indeed, the AT2 method introduced in [1] leverages a linear surrogate modeling approach (inspired by LIME) combined with attention weights, offering an enhanced attribution method compared to traditional attention-based approaches. We conducted an additional comparison to illustrate the relative performance:
>
>
> | Model      | Method     | CODE | BIO  | FP   | R-PRM | R-SEN | R-NUM | REF  | Avg. |
> |------------|------------|------|------|------|-------|-------|--------|------|------|
> | Llama-70B  | attention  | 34.0 | 0.7  | 5.0  | 39.3  | 1.1   | 58.4   | 0.1  | 19.8 |
> |            | lime       | 13.7 | 3.3  | 6.1  | 18.1  | 11.1  | 57.8   | 1.4  | 15.9 |
> |            | AT2       | 31.5 | 5.3  | 8.1  | 39.0  | 13.5  | 65.9   | 3.2  | 23.8 |
> |            | SAT (Ours) | 47.5 | 29 | 18.7 | 70.4 | 42.9 | 86.8 | 30.1 | 46.5 |
> | Olmo-7B    | attention  | 31.8 | 0.2  | 25.0 | 19.1  | 13.7  | 50.9   | 27.4 | 24.0 |
> |            | lime       | 21.7 | 0.9  | 11.7 | 19.8  | 6.0   | 56.6   | 17.0 | 19.1 |
> |            | AT2       | 27.2 | 4.5  | 17.4  | 22.3  | 18.9  | 60.2   | 28.1  | 25.5 |
> |            | SAT (Ours) | 39.7 | 17.3 | 57.6 | 29.4 | 30.1 | 75.5 | 48.4 | 42.5 |
>
>
> The AT2 method indeed exhibits stronger performance compared to conventional attention and LIME-based methods, effectively identifying locally relevant subsequences. However, since AT2 remains inherently focused on local-context attribution, it does not capture the broader reproducibility of hallucination-triggering subsequences across varied contexts. Our SAT framework, specifically designed for reproducibility-based analysis, demonstrates a notable advantage, underscoring the unique value of our approach.

---

> > ### Comment · Reviewer_4TsJ · 2025-08-03
> >
> > Thanks for the response. The explanation and additional experiments addressed my concerns. I trust the authors that the additional discussions and experiment results will be added to the paper. I will keep my score unchanged as 5 to accept this paper.

---

### Official Review · Reviewer_dLr7 · 2025-07-02

**Clarity:** 3
**Significance:** 3
**Originality:** 4
**Rating:** 5
**Confidence:** 3

**Summary:**

The paper proposes a framework for understanding LLM hallucinations through subsequence associations. The idea is that hallucinations occur when spurious input-output subsequence patterns learned during training outweigh faithful ones. They introduce a tracing algorithm (SAT) to identify causal subsequences that trigger hallucinations and evaluate it against standard attribution methods.

**Questions:**

1. The paper labels the "Lion King" → "Elton John" connection as an "irrelevant trigger" (line 36), but isn't this actually a valid semantic association since Elton John composed music for The Lion King? How do the authors distinguish between truly spurious associations versus legitimate connections that are simply contextually inappropriate?
2.  I would like to see the explanation for the questions raised in the weakness part.

**Ethical Concerns:**

["NO or VERY MINOR ethics concerns only"]

**Final Justification:**

I raised a my concerns regarding independence assumptions which was justified in the rebuttal. Also the concern regarding evaluation on Top-100 frequent hallucination patterns are well justified in the rebuttal. I support the acceptance of this paper for publication.

**Limitations:**

Yes

**Quality:**

4

**Strengths And Weaknesses:**

**Strengths**
1. The idea that hallucinations arise from  associations between input subsequences and outputs is intuitive and interesting
2. The theory fits well with how transformers work, especially the idea that each output token is influenced by a mix of earlier subsequences in the input.
3. The authors evaluate their SAT method thoroughly across 6 domains from the HALoGEN benchmark (CODE, REF, BIO, FP, R-PRM, R-SEN, R-NUM), two models (Olmo-7B and Llama-70B), and subsequence lengths at three ratios.
4. The finding that identified subsequence associations show strong correlation (ρ=0.72) with patterns in the Dolma training dataset suggests the method captures real learned patterns rather than random artifacts.

**Weaknesses**
1. The independence assumptions for subsequences (Proposition 2.3) doesn't hold in practice. In Natural language n-grams are highly dependent.

2. While the evaluation setup is generally strong, the decision to test SAT on only the top 100 most frequently occurring hallucinated subsequences per domain raises concerns about selection bias. These high-frequency patterns are likely to be overrepresented in the training data and may exhibit stronger, easier-to-trace associations. As a result, the method’s effectiveness may be overstated, and its generalizability to rarer or more complex hallucinations remains unclear.

Typos:
- line3: misisng fullstop  between diagnosis and However.

---

> ### Author Rebuttal · Authors · 2025-07-30
>
> **Opening Remark**: We thank the reviewer for the constructive and valuable feedback.
> We are honored that you found our framing of hallucinations as arising from associations between input subsequences and outputs both intuitive and insightful. We are equally pleased that the proposed theory aligns naturally with transformer mechanics—specifically, how each output token integrates influences from earlier input subsequences. Additionally, we appreciate your recognition of the thorough evaluation of our SAT method, as well as the strong correlation between our identified subsequence associations and patterns in the Dolma training data, which supports the validity of our approach.
> We have addressed the reviewer’s comments and concerns below.
>
> > **Independence Assumption in Proposition 2.3**
>
> We agree with the reviewer that, in practice, the independence assumption stated in Proposition 2.3 holds only under specific conditions. As noted at the beginning of Section 2.2, our intention is to provide an intuitive perspective of hallucination phenomena rather than a rigorous proof. Theoretical results typically require certain simplifying assumptions to deliver clear insights. Under these conditions, Proposition 2.3 effectively conveys that hallucination subsequences appearing in the model's output are influenced by the cumulative effects of key subsequence associations between input and output. This intuitive interpretation typically remains valid even when the independence assumption is relaxed.
>
> > **Evaluation on Top-100 Frequent Hallucination Patterns**
>
> Great question! The selection of the top-100 most frequent hallucination patterns serves a specific methodological purpose (targeting **reproducible hallucinations**) rather than introducing bias. Since all tracing methods in Table 2 are evaluated on identical test cases, the comparison remains fair and unbiased.
>
> Our approach targets reproducible hallucinations—a critical distinction in hallucination research. We observe that hallucinations generally fall into two categories:
>
> 1. **Reproducible hallucinations**: Consistent erroneous outputs that reliably appear across multiple samples with the same prompt context.
>
> 2. **Stochastic hallucinations**: Random errors resulting from low-probability token sampling during decoding, occurring perhaps once in 1,000 generations.
>
> Our framework is designed to analyze reproducible hallucinations, where the underlying causal mechanisms can be systematically studied. For stochastic hallucinations, the base reproduction rate approaches zero even with complete prompts, making it nearly impossible for any tracing method—including ours—to identify subsequences that meaningfully increase hallucination probability above baseline.
>
> By focusing on frequent, reproducible patterns, we ensure that our evaluation measures genuine causal attribution rather than random noise. This approach enables meaningful comparison of tracing effectiveness across methods and provides actionable insights for understanding systematic hallucination behaviors in LLMs. The high-frequency nature of these patterns reflects their systematic occurrence in model behavior, making them the most relevant targets for causal analysis and potential mitigation strategies.
>
>
> > **The Spurious Association versus Legitimate Association**
>
> Another great question! The distinction between spurious and legitimate associations indeed depends significantly on context. Consider the prompt, "The successful singer was born in New York. He never watched the Lion King. He is _____." Here, responding with "Elton John" constitutes a spurious association, as Elton John is neither an American singer nor plausibly described by the given context. This response thus reflects reliance on partial or misleading associations.
>
> Conversely, the prompt "The successful singer who composed songs for the movie Lion King is _____." correctly establishes a legitimate association with "Elton John." Hence, the legitimacy of an association is contextually determined rather than intrinsic. Our framework specifically addresses scenarios where contextually incomplete or misleading associations lead to hallucinations, distinguishing these from valid, contextually appropriate associations.

---

### Comment · Area_Chair_gq3U · 2025-08-04
**Reviewer-author discussion phase ends in 2 days**

Dear Reviewers,

Thank you for your reviews for this paper.

This is a kind reminder to please participate in the reviewer-author discussion phase, which ends in 2 days — by Tuesday, August 6 at 11:59 PM AoE. As the authors have responded to your reviews, we kindly ask that you read and engage with their responses, ask clarification questions as needed, and respond to help clarify key points before final decisions.

Your input during this phase is critical to ensuring a constructive and fair outcome.

Let us know if you have any questions or need assistance.

Warm regards,

AC

---

### Decision · Program_Chairs · 2025-09-17

**Decision:**

Accept (poster)

**Comment:**

This work proposes a framework that analyzes input token subsequences to systematically understand the hallucination behavior in large language models. They estimate the conditional probability of hallucinated tokens given each subsequence, identifying input triggers that induce the same hallucination and trace it back to the training data. They find that when spurious input-output subsequence patterns learned during training outweigh faithful ones, it can cause hallucinations. Experiments on halogen benchmark across multiple domains show that their proposed attribution method SAT substantially outperforms established attribution methods.

All reviewers note that both the theoretical and empirical observations are well-justified, thoroughly defined and the evaluations are comprehensive across multiple domains and models.

For the camera-ready version, I recommend that the authors add a broader impact section discussing the implications of their findings. I also encourage inclusion of the additional baselines and ablation studies (e.g., hyperparameter sensitivity) presented during the rebuttal, as requested by reviewers 4TsJ and eXJp. These additions would further strengthen the completeness, and practical value of the work.